# Counting in Small Transformers: The Delicate Interplay between Attention and Feed-Forward Layers

## Abstract

How do different architectural design choices influence the space of solutions that a transformer can implement and learn? How do different components interact with each other to shape the model's hypothesis space? We investigate these questions by characterizing the solutions simple transformer blocks can implement when challenged to solve the histogram task – counting the occurrences of each item in an input sequence from a fixed vocabulary. Despite its apparent simplicity, this task exhibits a rich phenomenology: our analysis reveals a strong inter-dependence between the model's predictive performance and the vocabulary and embedding sizes, the token-mixing mechanism and the capacity of the feed-forward block. In this work, we characterize two different counting strategies that small transformers can implement theoretically: relation-based and inventory-based counting, the latter being less efficient in computation and memory. The emergence of either strategy is heavily influenced by subtle synergies among hyperparameters and components, and depends on seemingly minor architectural tweaks like the inclusion of softmax in the attention mechanism. By introspecting models *trained* on the histogram task, we verify the formation of both mechanisms in practice. Our findings highlight that even in simple settings, slight variations in model design can cause significant changes to the solutions a transformer learns.

## 1 Introduction

Transformers are the key neural network behind many recent deep learning advances, most notably large language models (LLMs). Their success is partly due to their versatility in processing diverse data types, including text, images, and video, represented as sequences of tokens (Liu et al., 2021; Girdhar et al., 2019; Brown et al., 2020). While scale has been a key factor in unleashing the potential of these models, it is remarkable that their architecture still largely follows the same simple template of the original transformer model proposed by Vaswani et al. (2017). At its core, a single transformer block primarily alternates two basic components: the token-mixing attention mechanism and a standard fully connected multi-layer perceptron. At a high level, the attention mechanism mixes the tokens, while the multi-layer perceptron applies a nonlinear feature transformation identically to each token. Despite the widespread use of transformers, there is no clear consensus on the distinct roles of their components, how they interact, or if they can be substituted with alternative modules (Tolstikhin et al., 2021; Bozic et al., 2023; Gu & Dao, 2023). In particular, the specific contribution of each architectural element to the model's hypothesis space –the range of algorithms it can learn and implement in practice– remains opaque (Weiss et al., 2021; Delétang et al., 2023; Abbe et al., 2023; Ouellette et al., 2023).

In this work, we investigate this question from a mechanistic interpretability perspective (Cammarata et al., 2020; Olah et al., 2020; Elhage et al., 2021; Michaud et al., 2024; Ouellette et al., 2023) by considering the histogram task as a prototypical problem (Weiss et al., 2021). This task consists of predicting the number of appearances of each token in the input sequences processed by the model – counting. It encompasses two distinct fundamental algorithmic operations: comparison and aggregation. Despite its apparent simplicity, this task exhibits a rich phenomenology, allowing us to study the relative role of different architectural components and their impact on the final solutions implemented by the model in a *controlled* setting. To this end, we focus on models following

the architectural template of primitive transformer blocks, i.e. alternating a token-mixing attention mechanism and a multi-layer perceptron.

In our analysis, we provide explicit constructions (parameter configurations) for a range of such architectures reaching perfect accuracy in a model-dependent hyperparameter regime. In a subsequent step, we compare these algorithms with the performance and mechanistic behavior of models trained from data. Our findings reveal that this class of models is capable of implementing strikingly different solutions for the histogram task, with a strong dependence on the scale of the model's hyperparameters and the type of token-mixing mechanism utilized. Our main contributions are:

- We identify two main algorithmic strategies that can be used to solve the histogram task perfectly: *relation-* and *inventory-based counting*. Relation-based counting uses local pair-wise comparisons between tokens in a given sequence to obtain the number of occurrences conditioned on a given position. Inventory-based counting relies on the knowledge of the complete alphabet and counts the occurrences of all possible tokens to then extract the correct count for a given position.

- We show that the emergence of either mechanism during learning depends on the specifics of the architecture and the inductive bias it possesses in relation to the task. Relation-based counting is memory and compute-efficient as it can leverage an attention-like dot-product mixing mechanism for comparison operations. Inventory-based counting, instead, can be implemented based on an input-independent token-mixing mechanism. This weak inductive bias can be compensated via a feed-forward module with a large enough hidden layer that can memorize a lookup table to implement a comparison operation (inventory): the model-task misalignment can be closed at the cost of increased memory and compute requirements.

- When the embedding dimension is comparatively smaller than the size of the alphabet, we show that non-orthogonal embeddings can still result in some models attaining perfect accuracy. Due to the discrete nature of the counting task, near-orthogonal embeddings may not have a detrimental effect on prediction performance. Additionally, major gains are possible for the softmax operator and dot-product attention which together can remove noise stemming from linear dependence in a semantic, token-dependent manner. In this context, we also identify a curious regime where very small embedding dimensions, independent of the alphabet size, are in theory possible, but are never learned.

Section 2 provides the necessary background and notation. In Section 3 we describe our experimental setup, followed by our theoretical and experimental results[1] in Section 4. Section 5 discusses the related literature. Section 6 presents the limitations, conclusion and open questions of this work.

## 2 BACKGROUND AND NOTATION

**Architecture.** As inputs, we consider sequences of tokens $\mathbf{x} = (x_1, x_2, \cdots, x_L) \in \mathcal{T}^L$. Each token stems from the set $\mathcal{T} = \{1, \cdots, T\}$ of size $T$. The corresponding sequence of outputs $\mathbf{y} = (y_1, \cdots, y_L)$ has the same length as the input sequence, where each output token belongs to the output alphabet $\mathcal{C}$ of size $C$, i.e. $y_\ell \in \{1, ..., C\}$, with $C \leq L$. In this work, we analyze several 1-layer model architectures where a token-mixing mechanism is followed by a per-token feature transformation. This setup includes the case of a single transformer block where the dot-product attention mechanism is followed by a token-wise feed-forward network. Formally, we consider a model $F : \mathcal{T}^L \to \mathcal{C}^L$ defined for the positions $\ell = 1, \cdots, L$ as

$$F(\bar{\mathbf{x}})_\ell = \underset{c \in \{1, \cdots, C\}}{\arg\max} \ f(\bar{x}'_\ell)_c \ ; \ \ \bar{x}'_\ell = \bar{x}_\ell + [\mathbf{A}(\bar{\mathbf{x}})\bar{\mathbf{x}}]_\ell \tag{1}$$

with the token mixing matrix $\mathbf{A} : \mathbb{R}^{L \times d} \to \mathbb{R}^{L \times L}$ and the token-wise feature transformation $f : \mathbb{R}^d \to \mathbb{R}^C$. The embedding $\bar{\mathbf{x}} \in \mathbb{R}^{L \times d}$, where $\bar{x}_\ell$ denotes its $\ell$-th row, is obtained by passing the input sequence $\mathbf{x}$ into a standard embedding layer (learnable lookup-table) of dimension $d$. We refer to the embedding associated with token $t \in \mathcal{T}$ as $e_t \in \mathbb{R}^d$ or $e_{x_\ell} \in \mathbb{R}^d$ for the embedding of the token $x_\ell$ at position $\ell$. We do not include positional embeddings due to the inherent permutation equivariance of the histogram task. We refer to the vector $\bar{x}'_\ell$, for each position $\ell = 1, \cdots, L$, as the mixed token. Note that we assume that all operations in the network are executed with infinite precision. We comment when this becomes problematic.

---

[1]All results and code to reproduce them is available in the supplementary material.

**Token Mixing.** We consider two types of mixing mechanisms $\mathbf{A}$ with different activation functions. We refer to the case where the function $\mathbf{A}$ is constant in $\bar{\mathbf{x}}$ as *linear mixing* (`lin`), e.g.

$$\mathbf{A}_{\text{lin}}(\bar{\mathbf{x}}) = A, \qquad \mathbf{A}_{\text{lin+sftm}}(\bar{\mathbf{x}}) = \text{softmax}(A), \tag{2}$$

where $A \in \mathbb{R}^{L \times L}$ is a learnable matrix and the softmax operator is applied row-wise. The number of learnable parameters is therefore $L^2$. As an alternative mixing structure, which we refer to as *dot-product mixing* (`dot`), we consider the popular attention mechanism which constructs the matrix $\mathbf{A}$ to be explicitly dependent on the inputs, i.e.

$$\mathbf{A}_{\text{dot}}(\bar{\mathbf{x}}) = \frac{1}{\sqrt{d}}\bar{\mathbf{x}}W_Q W_K^T \bar{\mathbf{x}}^T, \qquad \mathbf{A}_{\text{dot+sftm}}(\bar{\mathbf{x}}) = \text{softmax}\left(\frac{1}{\sqrt{d}}\bar{\mathbf{x}}W_Q W_K^T \bar{\mathbf{x}}^T\right), \tag{3}$$

where $W_Q$ and $W_K$ are learnable $d \times d$ matrices, and the softmax function is applied row-wise. Note that, without loss of generality, we assume the value matrix to be the identity. The number of parameters for dot-product mixing is $2d^2$. In line with previous work (Weiss et al., 2021), for architectures employing the dot-product mixing, we also analyze models utilizing the so-called beginning-of-sequence (BOS) token. This special token, indicated with the symbol \$, is appended to the original input $\mathbf{x}$ resulting in a new sequence $\tilde{\mathbf{x}} = (\$, x_1, x_2, \cdots, x_L)$ of length $L+1$. We will refer to the architecture that includes the BOS token as `bos`.

**Feature Transformation.** The feature transformation is a single hidden layer perceptron with ReLU activations. The hidden layer is of dimension $p$. The function $f$ is applied identically to every mixed token $\bar{x}'_\ell$ for $\ell = 1, \cdots, L$, as:

$$f(\bar{x}'_\ell) = \text{ReLU}(\bar{x}'_\ell W_1 + b_1)W_2 + b_2 \tag{4}$$

where $f(\bar{x}'_\ell) : \mathbb{R}^d \to \mathbb{R}^C$ and where the weights have the appropriate dimensions to accommodate a hidden layer of size $p$, i.e. $W_1 \in \mathbb{R}^{d \times p}, b_1 \in \mathbb{R}^p, W_2 \in \mathbb{R}^{p \times C}$ and $b_1 \in \mathbb{R}^C$.

## 3 Experimental Setup

**Task and Dataset.** We consider a simple algorithmic task that is referred to as *histogram*: given a sequence of tokens, the goal is to return a sequence of the same length where each entry represents the number of times the corresponding input token appears in the entire sequence. For example, given $\mathbf{x} = [A, B, D, D, B, B]$, the output will be $\mathbf{y} = [1, 3, 2, 2, 3, 3]$. We define the count of a token $t$ in the sequence $\mathbf{x}$ at position $\ell$ as $hist_\mathbf{x}(\ell)$. In our experiments, we consider i.i.d. distributions of sequences of length $L$ from an input alphabet of size $T$, where $L \leq T$. Our sampling strategy relies on first sampling a set of partitions, and then assigning a token to each partition (see App. C for details). This allows for a close to uniform distribution over the values of $\mathbf{y}$.

**Models and Training.** We investigate the performance on the histogram task of the four different variants of the token mixing models described in Sec. 2, i.e. `lin` and `dot`, with or without the softmax (`+sftm`), where the token embeddings are jointly learned with the model parameters. Their relevant hyperparameters are the dimension of the embedded tokens $d$, and the hidden layer size $p$ of the feature transformation. Additionally, we consider the model `bos(+sftm)` where every input sequence is prefixed with the BOS token prior to entering a dot-product mixing layer (with softmax). Previous studies (Weiss et al., 2021; Kazemnejad et al., 2023) have demonstrated that transformer networks consistently attend to BOS tokens, despite their lack of semantic content, and we explore this point in our experiments.

All models are trained with Adam with a learning rate of $\nu = 10^{-3}$ on the cross-entropy loss for 500 epochs with a batch size of 32. We consider the online learning setting. For each batch we sample a new sequence of data from the generalize model. We compute the accuracy attained by each model based on a set of $3,000$ independent data samples, which covers a large range of all possible input sequences.

## 4 Learning Regimes in Counting

In order to understand the contributions of the different architectural components, we analyze the performance of the above-stated models with varying mixing mechanisms in different learning regimes characterized by the embedding dimension $d$ and the number of hidden neurons $p$ of the feed-forward module.

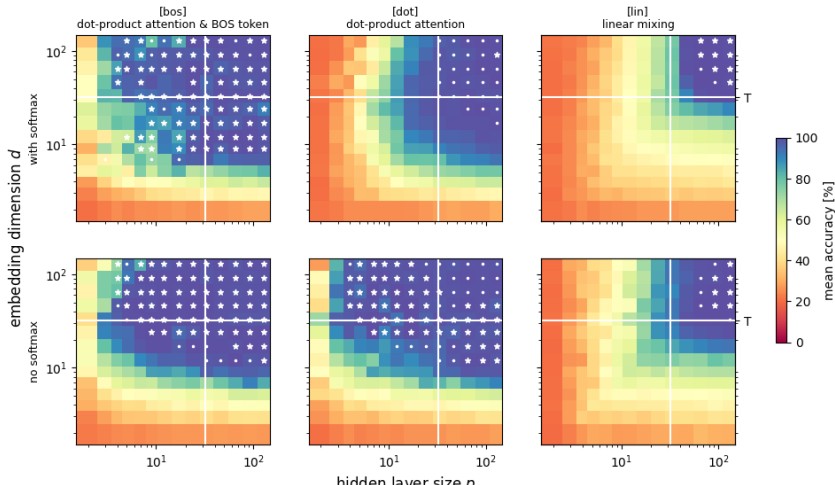

Figure 1: *Performance on the histogram task for different 1-layer transformer architectures.* Mean accuracy for varying embedding dimension $d$, hidden layer dimension $p$, for fixed $T = 32$ and $L = 10$ for the different token mixing mechanisms `dot`, `bos` and `lin`. *(Top)* Models with softmax; *(Bottom)* Models without softmax. Average over 5 runs for every $d, p \in \{1, 2, 3, 4, 6, 8, 12, 16, 23, 32, 45, 64, 91, 128\}$. Vertical and horizontal white lines indicate $p = T$ and $d = T$ respectively. White stars mark the parameter configurations, where a 100% accuracy configuration was found during training in at least one of the five runs. White dots mark the same for $\geq 99\%$ accuracy configurations.

Fig. 1 shows the accuracy attained by learned models for sequences of length $L = 10$ with $T = 32$ different input tokens. We observe that the models exhibit both high and low accuracy across various parameter regimes, with a strong dependence on the architecture. Fig. 2 further clarifies that the parameter efficiency under different architectures varies substantially.

To investigate the underlying mechanisms we devise theoretical constructions and mechanistic interpretations of the learned solutions. We delineate two regimes in each of the parameters: for the embedding dimension $d$ we distinguish the regime of non-orthogonal embeddings ($d < T$) and of possibly orthogonal embeddings ($d \geq T$). For hidden layer size $p$ we distinguish the regime where models can sense only a constant number of directions/features ($p = 1$) or one scaling as the alphabet size ($p = T$).

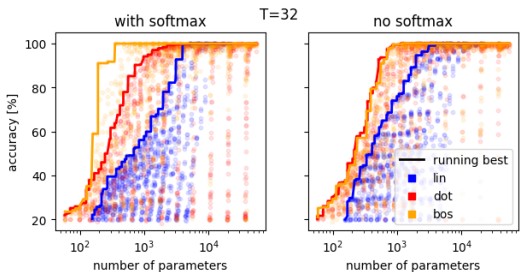

Figure 2: *Accuracy vs. Parameter count.* The data is the same as generated for Fig. 1, every data point is a single experiment and we show the convex hull in solid lines.

### 4.1 $d \geq T$: ORTHOGONAL TOKEN EMBEDDINGS ARE SEPARABLE

When the model dimension $d$ is at least as big as the number of tokens $T$, tokens can be represented by embeddings that are mutually orthogonal to one another. Assuming all tokens $t \in \mathcal{T}$ have such mutually orthogonal embeddings $e_t \in \mathbb{R}^d$ with a norm of 1, the overlap is $\langle e_s, e_t \rangle = 0$ for distinct tokens $t \neq s$ and it is 1 when $t = s$. In such a scenario, a linear combination of token embeddings preserves magnitudal - count - information about single tokens. By leveraging knowledge about the embeddings of the alphabet, a weighted sum of tokens, denoted as $e' = \sum_{t \in \mathcal{T}} \alpha_t e_t$, can be broken down into the original tokens using projections on the original token embeddings, where $\alpha_t = \langle e_t, e' \rangle / \|e_t\|_2^2$.

In the following, we use this property to *theoretically* construct the weights for all models that solves the task when $d \geq T$. Remarkably, the constructions require different number of hidden neurons $p$

depending on the mixing mechanism. This demonstrates the interplay of the mixing layer and the feature transform: for some mixing mechanisms, the latter needs to implement *inventory-based counting* (IC) (requiring $p \geq T$), and for others, *relation-based counting* (RC) (where $p \geq 1$ is sufficient).

### 4.1.1 RELATION-BASED COUNTING: LEVERAGING DOT-PRODUCT MIXING

When an extra beginning-of-sequence token $t_{\text{BOS}}$ is available in bos, it can be used as a to extract information about a token's count $hist_{\mathbf{x}}(\ell)$ in the attention layer of the network through its attention score Kazemnejad et al. (2023). In the literature, the beginning (or end) of sequence tokens have been linked to model-internal computations, such as counting. In Weiss et al. (2021), it is shown that the RASP language can solve the histogram task with one layer and one attention head. We confirm empirically that bos and bos+sftm reach (close to) 100% accuracy whenever $d > T$, and we verify that a relation-based counting algorithm can be theoretically implemented in these two architectures by construction.

**Proposition 1** (RC with BOS token). *For bos and bos+sftm and a given $L \geq 2$, there each exists a configuration of weights that solves the histogram task at 100% accuracy, given that $d \geq T > 2$ and $p = 1$.*

We prove this by construction in App. A.2.3-A.2.2 and we provide the intuition of the proof in the following. For bos we set the $t_{\text{BOS}}$ embedding to $e_{\text{BOS}} = \sum_{t \in \mathcal{T}} e_t$ and take the mutually orthogonal token embeddings $e_t$ to have norm 1. Assuming that $t_{\text{BOS}}$ is at the first position of the sequence of now length $L + 1$, a simple dot-product operation in the attention mechanism (with $Q, K = d^{\frac{1}{4}} \mathbb{I}_d$) will lead to an attention matrix with entries:

$$a_{\ell m} = \begin{cases} T & \text{if } \ell = m = 1 \\ 1 & \text{if } (\ell > 1, m = 1) \text{ or } (\ell, m > 1, x_\ell = x_m) \\ 0 & \text{if } \ell, m > 1, x_\ell \neq x_m \end{cases}.$$

Projecting the mixed token $\bar{x}'_\ell$ onto the $t_{\text{BOS}}$ we obtain $\langle \bar{x}'_\ell, e_{\text{BOS}} \rangle = T + hist_{\mathbf{x}}(\ell) + 1$, i.e. $e_{\text{BOS}}$ is the single relevant direction for the prediction. Its magnitude relates linearly to $hist_{\mathbf{x}}(\ell)$. A *single* hidden neuron $p = 1$ suffices and the output layer can transfer the count into a categorical representation. For bos+sftm one needs to further account for the non-linearity of the softmax as described in App. A.2.2.

In the learned models, some instances in the given regime indeed achieve 100% accuracy. While their weights do not correspond exactly to the relation-based counting algorithm described previously, they exhibit similar properties. In Fig. 3, we show for bos+sftm, that $t_{\text{BOS}}$ indeed plays a special role in the *learned* model: in the attention matrix its activation can be interpreted as a proxy for the number of occurrences of $x_\ell$, as it has different values for tokens that occur a different amount of times. Other entries of the attention matrix are comparatively low when the compared tokens are the same and high when they are different. The comparison operation naturally provided by the dot-product allows the model to extract the count of the same tokens, for each token in the sequence. We also show in Fig. 3 how the presence of the $t_{\text{BOS}}$ determines the final prediction through the application of $f$.

Surprisingly, the dot model (without the softmax) reaches a an empirical performance comparable to bos in the regime $d \geq T$ and $p = 1$, even though it does not have an extra token available.

**Proposition 2** (RC with tagged embeddings). *For dot and a given $L, T > 2$, there exists a configuration of weights that solves the histogram task at 100% accuracy, given that $d \geq T > 2$ and $p = 1$.*

We prove this in App. A.2.1. Intuitively, the construction uses a single common direction $e_{\text{cnt}}$ that is added to the otherwise mutually orthogonal token embeddings. A dot-product mixing then leads to $a_{\ell m} = a_{\neq} > 0$ when $x_\ell$ is different from $x_m$, and $a_{\ell m} = a_{=} > 0$ when tokens are the same. Then, the number of counts can be easily extracted from the dot-product $\langle e_{\text{cnt}}, \bar{x}'_\ell \rangle$ of the counting token with the mixed token $\bar{x}'_\ell$, i.e. $\langle e_{\text{cnt}}, \bar{x}'_\ell \rangle \propto 1 + hist_{\mathbf{x}}(\ell) a_{=} + (L - hist_{\mathbf{x}}(\ell)) a_{\neq}$. We can, therefore, obtain a perfect accuracy implementation in the regime where $d \geq T$ with only a single hidden neuron. This is in line with the observed empirical performance by dot even without access to a BOS token.

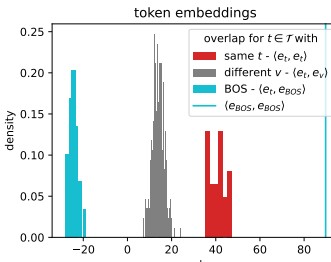 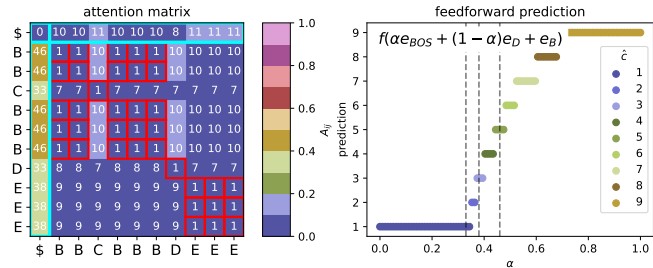

Figure 3: *Relation-based counting with* `bos+sftm` *($T = 32, L = 10, p = 2, d = 45$).* This model achieves 99.9% accuracy. It was selected as the best model from all our experiments with $p = 2$. *(Left)* The tokens overlap (cosine similarity) with the same tokens (red), different tokens (grey) and the BOS (light blue) all concentrate on different values. *(Middle)* This is reflected in the attention matrix after the application of the row-wise softmax. The $t_{\text{BOS}}$ ('\$') in the first column $a_{\ell,0}$ becomes a proxy for the count of $x_\ell$. *(Right)* To demonstrate that the feedforward network is only sensitive to this direction, we show its count predictions for a mix of tokens $\alpha e_{\text{BOS}} + (1 - \alpha)e_D + e_B$, where the contribution $\alpha$ of the BOS token is varied and $D, B$ are two specific elements of the alphabet $\mathcal{T}$. The same experiment is repeated for different elements of the alphabet in App. D.6. We mark the $a_{\ell,0}$ obtained from the left as vertical lines, the prediction is correct for all counts independent of the precise token.

**Dot-product attention with softmax fails to implement relation-based counting.** Since the dot-product mechanism can naturally be used in relation-based counting, one might expect the `dot+sftm` model to implement the same mechanism. However, and maybe surprisingly so, we empirically observe a marked difference between `dot` and `dot+sftm` in Fig. 1. `dot` only starts performing close to 100% accuracy when both the model dimension $d$ and the number of hidden neurons $p$ are larger than the number of tokens $T$. To understand why it fails to learn for $p = 1$, we show the attention matrix of `dot+sftm` in Fig. 4. Notably, it is based on the semantics, as $(\mathbf{A}_{\text{dot+sftm}})_{\ell m}$ is higher when $x_\ell = x_m$ than otherwise. However, the normalization effect of the softmax activation prevents the development of a meaningful counter subspace that is needed in the relation-based algorithm. As a result of normalization, the attention scores are $\sum_m a_{\ell m} = 1$, so any direction present in all tokens (and by the symmetry of the task, it would need to be present in all tokens) would be uninformative after the token mixing – its weight would be one regardless of the input sequence and would therefore not carry information about the count. Before, the model `bos+sftm` circumvented this problem by adding the extra token with a special functionality that does not need to be counted. Because this is not possible for `dot+sftm`, the architecture fails to perform well for $p = 1$ – it now needs to measure more than one direction in the feed-forward module.

In the following, we show that a solution of the histogram task can still be achieved through an inventory-based counting algorithm with $p \geq T$. We detail this in the following section, for the example of `lin`. The statement for `dot+sftm` is given in App. A.3.

### 4.1.2 INVENTORY-BASED COUNTING: MEMORIZATION IN THE FEED-FORWARD LAYER

When the feed-forward hidden layer has one neuron for each distinct token available in the alphabet, it can detect as many directions. This allows the feed-forward layer to extract the information of any token direction separately and thereby implement a custom comparison operation that works for all of the tokens in the alphabet. While this is less parameter efficient and requires memorizing the complete alphabet, it enables the model to solve the task.

**Proposition 3** (IC with memorization in the feed-forward layer). *For* `lin` *and* `lin+sftm` *and a given $L, T > 2$ there exists a configuration of weights which solves the histogram task for $p \geq T$ and $d \geq T$.*

We describe examples of such constructions in App. A.3.1 and A.3.2. Again, several solutions exist due to symmetries, and in the following we give an intuition for one of them.

In the linear mixing layer $\mathbf{A}_{\text{lin}}$ we set a constant value $a = 1/L$ so that the result of the mixing is simply a position-independent linear combination of the input. The count $hist_{\mathbf{x}}(\ell)$ can be extracted

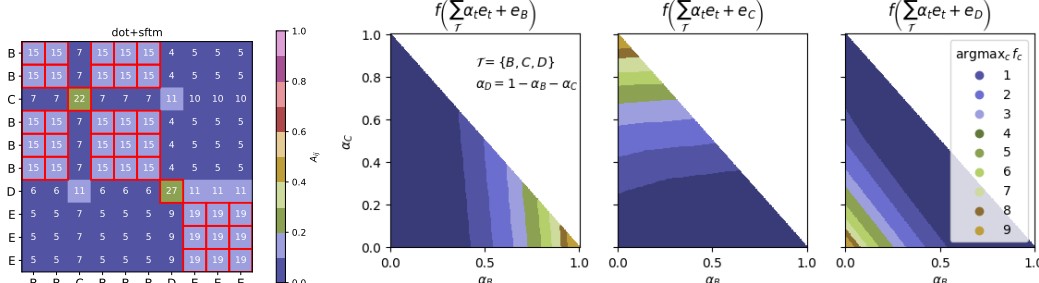

Figure 4: *Inventory-based counting with* `dot+sftm` $(T = 32, L = 10, p = 32, d = 32)$. This model achieves $99.47\%$ accuracy. *(Left)* The attention matrix for a given sequence differentiates between similar and different tokens. However in this case, any counting direction that could emerge in token space is evidently not usable, as $p \geq T$ is required (see Fig. 1). *(Right)* This is reflected in the output from the feature transformation $f$, shown here for a linear combination of three different tokens from the alphabet, $B, C, D$. The prediction strongly depends on the coefficient $\alpha_t$ associated with the token $t$ present in the residual connection and only weakly on the others. The non-linear scaling of the decision boundaries is due to the softmax activation function.

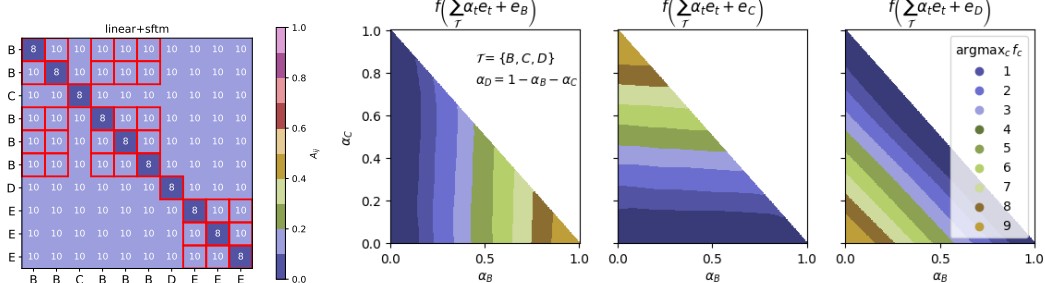

Figure 5: *Inventory-based counting with* `lin+sftm` $(T = 64, L = 10, p = 128, d = 128)$. The model achieves $99.97\%$ accuracy. *(Left)* Attention matrix learned by Adam, which is constant in the input sequence $\mathbf{x}$. The different score on the diagonal assigns a different weight to the token at the current position $\ell$ than to all other tokens. *(Right)* Predictions on an artificial mix of learned embeddings for the three tokens $B, C$ and $D$. The prediction depends on the token in the residual connection, but is largely independent of the presence of other tokens in the mixing. This indicates that $f$ projects the mixed token onto the alphabet $\mathcal{T}$ and is able to extract tokens due to orthogonality.

after the residual connection where we add $\bar{x}'_\ell = \bar{x}_\ell + e_{x_\ell}$. By setting the columns of the matrix $(W_1)_t = e_t$ we can extract the count information up to the factor $1/a$

$$hist_\mathbf{x}(\ell) = \frac{1}{a} \sum_{t \in \mathcal{T}} \text{ReLU}\left(\langle \bar{x}'_\ell, (W_1)_t \rangle - 1\right) = \frac{1}{a} \sum_{t \in \mathcal{T}} \text{ReLU}\left(\langle \bar{x}'_\ell, e_t \rangle - 1\right) = \frac{1}{a} \langle \bar{x}_\ell, e_{x_\ell} \rangle$$

Note that, due to the $-1$ bias term, only the hidden neuron for token $(W_1)_t = e_t = x_\ell$ that occurs in the residual connection has a non-zero activation. The output layer $W_2$ can then be designed to activate the correct output vector corresponding to the count $hist_\mathbf{x}(\ell)$ (see App. A.4). Since $a \in [0, 1]$ and $\sum_{m=1}^L a_{\ell m} = 1$ the same procedure can be implemented by a matrix which is passed through the softmax operator for `lin+sftm`. In practice, in this construction the feed-forward module is correlated with the complete alphabet, acting as an inventory, or look-up table.

In Fig. 5, we inspect the attention matrix $\mathbf{A}_{\text{lin}}$ and the feature transformation $f$ which is *learned* for `lin+sftm` in the regime where $p \sim T \sim d$. The mixing has an off-diagonal of $\sim 0.11$ and a diagonal of $\sim 0.08$. Feeding the feature transformation $f$ with a weighted combination of 3 tokens, $B, C, D$, we observe that the final prediction of the network depends mainly on the coefficient $\alpha_t$ corresponding to the token embedding fed through the residual connection. Notably, this behavior is

close to Fig. 4 (right) and suggests that the feature transformation must have encoded the information of the token embedding in its weights, hence requiring at least $p = T$ hidden neurons.

**Superpositioned and selective implementations.** Some of the models capabilities include one another. For example, the models that can implement relation-based counting for $p = 1$ can also implement the solutions for inventory-based counting for $p \geq T$. It is unclear, whether the memory-intensive solution is preferred when the memory is available, or if the efficient solution is learned nonetheless. Curiously, in Fig. 1, we observe that the model `dot` (which is capable of RC) witnesses a very slight decrease in maximal learned performance from $100\%$ accuracy to $99\%$ despite its capacity being *increased* to $p = T$ when inventory-based counting can in principle be implemented. In App. D.7 we investigate the singular value decomposition of $W_1$, for learned models with $\geq 99\%$ accuracy and $p, d \geq T$. We find that the largest $T = 32$ singular values are larger than the surplus singular values when $p > T$ for models that can implement only IC. This behavior is less pronounced for models that can implement RC, where the largest singular value is often relatively much larger than the following $T = 32$, but still show a small dip after the $T = 32$ singular values. Understanding which algorithm is implemented in this regime, or if it is a superposition of the two, thus requires further investigation.

### 4.2 $d < T$: Non-orthogonal embeddings and the discrete nature of counting

The scenario where $d < T$ fundamentally differs from the one explored in Section 4.1 because the embeddings for different tokens can no longer be mutually orthogonal. Some token pairs then have a non-zero overlap due to their linear dependence, causing the mixing of tokens to entangle count information across different directions in the embedding space. This phenomenon is illustrated for `dot` in Fig. 6, where learned models with smaller $d$ tend to overcount items in the input, and observe a less spread distribution of overlaps. Nevertheless in Fig. 1 we observe a number of results that empirically show almost perfect accuracy solutions with $d < T$ both for models with RC or IC.

Indeed, the discrete nature of the histogram task, i.e. the fact that every token can only be mapped to $L$ distinct counts, makes the prediction inherently more robust to the effect of noise stemming from entangled embeddings. This concept is illustrated in Fig. 7 in App. A.4 for the `dot+sftm` model. As long as the value of the logits in the final output layer falls within the margin between two counts the model still solves the task with perfect accuracy. The relative size of this margin decreases when $L$ is increased, making the task harder when more classes need to be distinguished.

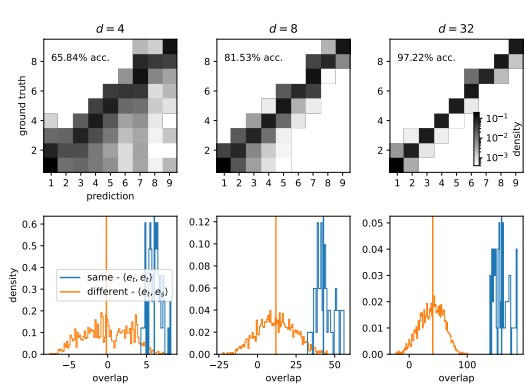

Figure 6: *Introspecting the Regime with Entangled Embeddings with* `dot` *($T = 64, L = 10, p = 128$). We show examples of* `dot` *for $T = 64, L = 10, p = 128$ for varying the model dimension $d$. (Top) The confusion matrix of ground truth and predicted counts. (Bottom) The overlap distribution between same and different token embeddings.*

In the following, we link concepts on optimally placing decision boundaries for noise robustness to a characterization of this entanglement noise, measured by the mutual coherence of the token embedding set (i.e., the maximum absolute overlap between pairs of distinct embeddings). The mutual coherence of a set of $T$ vectors of dimension $d$ is lower bounded by the Welch bound (Welch, 1974). This gives a means to understand the size of $d$ a given task with $T, L$ requires at least.

**Proposition 4** (Robustness via bounded mutual coherence). *Given $L \geq 5, T \geq 2$ and assuming that the Welch bound is attained for a given $T, d$, there exists a construction that solves the histogram task with*

*(`lin`, `lin+sftm`; $p = T$):* $\left\lceil \frac{T(2L-3)^2}{T-1+(2L-3)^2} \right\rceil \leq d$,

*(`dot`, `bos`; $p = 1$):* $\left\lceil \frac{T(2L-3)^2}{T-1+(2L-3)^2} \right\rceil + 1 \leq d$,

*(`dot`, `bos`; $p = T$):* $\left\lceil \frac{T(L-1)}{T-1+(L-1)} \right\rceil \leq d$.

We provide additional background and the proofs in App. B.2. The idea is to use constructions analogous to the RC and IC with orthogonal embeddings, while keeping track on how the errors of non-zero overlaps between pairs of different embeddings propagate through the model. For a given $L$ and $T$ this provides an upper bound on the maximal mutual coherence that is tolerated for a perfect solution. This can be connected to the dimensionality $d$ via the Welch bound. Evaluating the bounds for the setting in Fig. 1, we obtain, in the order of the above list, $d \geq 29, 30, 7$. Generally it is hard to generate matrices that attain the Welch bound and manually we did not succeed to find them for $d = 29, 30$. However we can indeed create an explicit construction a for dot and $p = T$ which attains $d = 12$, as provided in the supplementary code and in correspondence with Fig. 1. While this bound does not reach the $d$ as indicated by the Welch bound, the mutual coherence of the embedding matrix we use is close to the maximally allowed value of $\mathcal{M} = 0.299 < 1/3$.

The previous results apply specifically to models without the softmax operator in the token mixing step – models with this non-linearity can be more robust and attain even smaller $d$, as clearly visible in Fig. 1. The idea is that a softmax function with a high enough inverse temperature can non-linearly scale down the attention scores for different token pairs relative to those of the same tokens. Thereby, the noise introduced in the dot-product layer through pairs of different embeddings becomes *arbitrarily* close to zero after applying the softmax.

**Proposition 5** (Robustness via softmax error-reduction). *Given $T, L > 2$, there exist weight configurations that solve the histogram task for the parameter combinations (bos+sftm; $p = 1$) and (dot+sftm; $p = T$) with $\lceil \log_2(T+1) \rceil + 2 \leq d$.*

Put simply, this construction requires that there are token embeddings for $t, s = 1, \ldots, T$ and $s \neq t$ with $\epsilon > 0$ such that

$$\langle e_t, e_t \rangle = 1 \quad \text{and} \quad \langle e_t, e_s \rangle < 1 - \epsilon \,. \tag{5}$$

This is fulfilled when every token is the binary encoding of its value, modulo minor modifications due to the RC mechanism for bos+sftm. Setting the softmax temperature high enough as a function of $L$ allows for the contributions from non-equal tokens to be decreased relative to the ones of same tokens. Evaluating this function for Fig. 1, we obtain $d = 7$, which closely corresponds to the most parameter efficient solutions of the histogram task that we observe. As $L$ grows, we require stronger concentration from the softmax by adjusting its temperature. Since real-world networks execute finite computations, computational instabilities or collapses might occur. It is therefore not clear that this correspondence will hold for all values of $L$.

In App. B.3.1 we show that this bound can be even further improved for bos+sftm to a constant $d = 4$, but at the cost of increasing the temperature further as a function of $T$, in addition to $L$. This might be the reason why we do not observe any learned solutions of the histogram task in this regime.

## 5 RELATED WORK

**Mechanistic Interpretability and Counting.** The emergence of algorithmic capabilities in transformers (Olsson et al., 2022; Power et al., 2022) has led to numerous investigations aimed at reverse-engineering trained models into human-understandable mechanisms (Zhong et al., 2023; Nanda et al., 2023; Quirke & Barez, 2024). Previous studies have investigated a variety of histogram tasks and the mechanisms behind them (Gould et al., 2023; Chollet et al., 2020; Ouellette et al., 2023; Cui et al., 2024). In our work, we consider the histogram task introduced within the context of the RASP(-L) programming language (Weiss et al., 2021; Abbe et al., 2023). Weiss et al. (2021) predict that single layer transformers with one head require an additional BOS token as a scratchpad (Nye et al., 2021) to be able to solve the task. However, we find that the task does not necessarily require the BOS token and we give explicit constructions for several of such one-layer architectures. Our main focus is the interpretation of the hyperparameter scaling of several distinct models in relation to their performance and explicit constructions of different algorithms, similar to the studies in Zhong et al. (2023); Quirke & Barez (2024). We give precise theoretical conditions on the model configurations that lead to perfect explicit constructions. While many works in this area focus on causal interventions (Vig et al., 2020; Meng et al., 2023) to understand the computational mechanisms of models or assign relevance scores to their components (nostalgebraist, 2020; Elhage et al., 2021), our approach primarily involves gaining insights through direct introspection of the model's components.

**Memorization and Feed-forward Layers.** The role of feed-forward layers as memorization modules has been investigated in the context of factual recall for language models (Geva et al., 2021;

Meng et al., 2023; Chughtai et al., 2024). Henighan et al. (2023) study a double decent phenomenon where the purpose of the feed-forward layer transitions from storing data points to discovering generalizing features as a function of increasing training data diversity (Raventos et al., 2023). In the histogram task, we observe a similar phenomenon as a function of the architecture: the feed-forward layer acts either as a look-up table or a feature detector for a single direction in embedding space – the counting subspace.

**Aligning Algorithm and Architecture.** While theoretical work has outlined the computational capacity of a range of (autoregressive) neural networks (Weiss et al., 2021; Yun et al., 2019; Delétang et al., 2023; Liu et al., 2023), hallucinations and failure modes on seemingly trivial tasks in real-world transformers are the rule rather than an exception. Dziri et al. (2023) postulate that this may be due to a misalignment between the computational graph of a model and the task itself. In this work, we show that subtle differences in components such as the mixing type and layer width play a crucial role in terms of algorithmic alignment. Previous work discovered evidence for the superposition of different computational graphs in a single model (Elhage et al., 2022) – we complement this analysis with a toy model that is able to disentangle non-orthogonal, hence superimposed, embedding directions in some parameter regimes.

## 6 DISCUSSION & CONCLUSION

**Limitations.** Similar to other works in mechanistic interpretability (Zhong et al., 2023), we focus on 1-layer transformers as a simplified model for modern transformers. Our models are not autoregressive and do not account for the impact of causal masks or positional encodings. While more complex models could lead to more intricate interdependencies between the components, potentially limiting the applicability of our findings to such architectures, it seems plausible that similar vector arithmetic could emerge in subspaces of large transformers (Gould et al., 2023; Engels et al., 2024). Given its specificity, it is unclear if and how similar memory-architecture phenomena would emerge for different simple tasks (e.g. sorting or lookup).

**Summary.** We study how different components of simple transformer models contribute to the emergence of different solutions to the histogram task. Our analysis shows that the parameter regimes where solving the histogram task is feasible for these models is influenced by the choice of the mixing mechanism and its inter-dependency with the feed-forward transformation, as well as the softmax activation function in the attention mechanism. We identify two distinct algorithmic approaches that 1-layer transformers can utilize to solve the histogram task: relation-based counting and inventory-based counting. The relation-based method employs a dot product mixing mechanism combined with a low-capacity feed-forward transformation and relies on the presence of an appropriate counter direction within the token embedding space. In contrast, the inventory-based method involves memorizing the token embeddings within the feed-forward module's weights, thus requiring more parameters. By characterizing the feasibility regimes of these mechanisms in the phase space defined by the embedding dimension $d$ and the hidden dimension $p$ of the feed-forward module, we confirm that learned models converge to solutions resembling these mechanisms. In certain regimes, both strategies can potentially be implemented, and our experiments indicate that some learned models exhibit features of superimposed algorithmic mechanisms. In the regime where the embedding dimension $d$ is smaller than the alphabet size $T$, tokens cannot form an orthogonal basis and solve the task directly via a linear projection. Despite this, we find that the considered models exhibit different levels of robustness to the noise stemming from non-orthogonality. Our analysis precisely characterizes how different models cope with this aspect and identifies less stringent feasibility regimes in terms of the embedding dimension. In particular, we find that the softmax activation can be very effective in minimizing the effective similarity between distinct tokens after a comparison opearation through the attention layer, hence reducing the impact of non-orthogonality. This is particularly relevant to real world models, where the alphabet size is usually much larger than the model dimension.

**Future Directions.** At this moment, examples for hallucinations and failures of LLM's are as numerous as their success stories. Even though we only analyze the feasibility regime of a single task, this small example already exhibits a rich phenomenology. It shows that a number of subtle modifications to a models architecture can influence its predictive power drastically. The prime example is the softmax function which becomes a curse or a blessing depending on slight differences in the setup. We expect that similar mechanistic investigations at or close to the regimes where models start failing will be extremely useful to understand how and why models fail in sometimes puzzling manners.

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

APPENDICES

# A EXPLICIT CONSTRUCTIONS FOR ORTHOGONAL EMBEDDINGS $d = T$

## A.1 OVERVIEW

In the parameter regime where $d \geq T$ there is always an orthonormal basis of size $T$ in $\mathbb{R}^d$, these explicit constructions give the correct prediction for all input token sequences. For all the models we describe below, we define the sum of the hidden layer neurons as:

$$\gamma_\ell = \sum_{i=1}^{p} \text{ReLU}(z_{\ell,i}) = \sum_{i=1}^{p} \text{ReLU}(W_1 \bar{x}'_\ell + b_1)_i \tag{6}$$

In many cases, a simple linear regression can map the scalar $\gamma_\ell$ to the correct count of tokens $x_\ell$, and we describe how to achieve this mapping to the classification problem in Section A.4.

In the following, we characterize which parameters $W_1, b_1$ in equation 6 allow for a correct mapping in each mechanism. Importantly, the architecture exhibits numerous symmetries due to the feed-forward ReLU network (Petzka et al., 2020). To demonstrate feasibility, we select one specific implementation. In the main text we observe that there is no one-to-one correspondence between our explicit constructions and the learned weights, even though both functions achieve the same perfect accuracy. Throughout, unless otherwise specified, we assume that $E \in \mathbb{R}^{d \times T}$ is an orthonormal basis of $\mathbb{R}^d$, which we will use to create different forms of token embeddings.

---

The supplementary code at https://github.com/to-be-deanonymized contains executable `pytorch` models that have the weight configurations that are used to prove Propositions 2-3 and 6, which allows one to test the devised weight configurations for fixed $T, L, d$ in practice.

---

## A.2 RELATION-BASED COUNTING

### A.2.1 (DOT; $p = 1$)

*Proof of Proposition 2.* We set $T = d > 2$ with $L \geq 2$ and $p = 1$. We choose the embeddings of the tokens of the `dot` model as

$$e_t = \tilde{e}_t + \tilde{e}_{\text{cnt}} \quad \forall t = 1, \ldots T \tag{7}$$

where the set $E = \{\tilde{e}_t\}_{t=1}^T$ is an orthonormal basis of an arbitrary but fixed $T$-dimensional subspace of $\mathbb{R}^d$, and $\tilde{e}_{\text{cnt}} = \sum_{t=1}^T \tilde{e}_t$. The key and query matrix are set to the scaled identity $W_K = W_Q = d^{1/4} I_d$ and hence the mixing layer $\mathbf{A}_{\text{dot}}$ can be viewed as carrying out the unmodified dot-product operation between all pairs of tokens. The first layer weights $W_1, b_1 \in \mathbb{R}^d$ can be fixed as

$$W_1 = \tilde{e}_{\text{cnt}}/(T+1); \quad b_1 = -(1 + L(T+2)), \tag{8}$$

and the second layer weights $W_2, b_2 \in \mathbb{R}^L$ follow the recursion

$$(W_2)_1 = -1 + \frac{1}{L+1}, \quad (b_2)_1 = 0; \tag{9}$$

$$(W_2)_\ell = -1 + \frac{\ell}{L+1}, \quad (b_2)_\ell = ((W_2)_{\ell-1} - (W_2)_\ell)(\ell - 0.5) + b_{\ell-1}, \quad \forall \ell = 2, \ldots, L. \tag{10}$$

Given these parameters, it holds that for tokens $1 \leq t, s \leq T$ their dot-product is

$$\langle e_t, e_s \rangle = \begin{cases} 2 + T & \text{if } t \neq s, \\ 3 + T & \text{if } t = s. \end{cases} \tag{11}$$

Because of our choice of the query and key matrices, it directly follows that for tokens $x_\ell, x_m$ at positions $\ell$ and $m$ from a given sequence $\mathbf{x}$, their attention score is

$$a_{\ell m} = \begin{cases} 2 + T & \text{if } x_\ell \neq x_m, \\ 3 + T & \text{if } x_\ell = x_m. \end{cases} \tag{12}$$

Hence, the mixed token after applying the residual connection is

$$\bar{x}'_\ell = \bar{x}_\ell + \sum_{m: x_\ell = x_m} (T+3)\bar{x}_m + \sum_{m: x_\ell \neq x_m} (T+2)\bar{x}_m \tag{13}$$

so that computing

$$\bar{x}'_\ell W_1 = \left\langle \bar{x}'_\ell, \frac{\tilde{e}_{\text{cnt}}}{1+T} \right\rangle \tag{14}$$

$$= \left\langle \bar{x}_\ell, \frac{\tilde{e}_{\text{cnt}}}{1+T} \right\rangle + \sum_{m:x_\ell=x_m} (T+3) \left\langle \bar{x}_m, \frac{\tilde{e}_{\text{cnt}}}{1+T} \right\rangle + \sum_{m:x_\ell \neq x_m} (T+2) \left\langle \bar{x}_m, \frac{\tilde{e}_{\text{cnt}}}{1+T} \right\rangle \tag{15}$$

$$= 1 + hist_{\mathbf{x}}(\ell)(T+3) + (L - hist_{\mathbf{x}}(\ell))(T+2) \tag{16}$$

$$= hist_{\mathbf{x}}(\ell) + 1 + L(T+2). \tag{17}$$

Then the single hidden unit has the value $\gamma_\ell = \text{ReLU}(\bar{x}'_\ell W_1 + b_1) = hist_{\mathbf{x}}(\ell)$. It is easy to show (analogous to Fig. 7) that the output logits $c = \gamma_\ell W_2 + b_2$ with $c \in \mathbb{R}^L$, correctly identify the count for integer values $x \in [1, \ldots, L]$. This is because we constructed our recursion such that at a given input $x = \ell$ we have that $(W_2)_\ell(\ell - 0.5) + (b_2)_\ell = (W_2)_{\ell-1}(\ell - 0.5) + (b_2)_{\ell-1}$ and $(W_2)_\ell > (W_2)_{\ell-1}$, so it holds that

$$\underset{i=1\ldots L}{\arg\max}\, c_i(y) = \begin{cases} 1 & y = 1 \\ 2 & y = 2 \\ \cdots & \\ L & y = L, \end{cases} \tag{18}$$

which gives the correct classification output for all possible inputs, and hence solves the histogram task at 100% accuracy. □

Note, however, that this weight configuration is only one example, and some symmetries in the model can lead to different but also 100% correct algorithms. This is especially important as we compare the regime outlined in the Theorem with the weight configurations learned.

### A.2.2 (BOS+SFTM; $p = 1$)

*Proof of Proposition 1 for* bos+sftm. We set $T = d > 2$ with $L \geq 2$ and $p = 1$ and consider the model dot+sftm. Note that in this model every sequence $\mathbf{x}$ is prefixed with $t_{\text{BOS}}$ before it is fed into the embedding and then the mixing layer. Again we use mutually orthogonal embeddings. $E = \{\tilde{e}_t\}_{t=1}^T$ is an orthonormal basis of an arbitrary but fixed $T$-dimensional subspace of $\mathbb{R}^d$, and $\tilde{e}_{\text{cnt}} = \sum_{t=1}^T \tilde{e}_t$. We set $e_{\text{BOS}} = \sum_{t=1}^T E_t$, where $e_t = E_t$ and the latter is a column of $E$. Analogous to the background token from Proposition 1 there is only one direction $p = 1$ to detect in the feedforward model, so we set

$$W_1 = e_{\text{BOS}}; \quad b_1 = -1. \tag{19}$$

For a given token $x_\ell$ we have that in the dot-product mechanism $\langle e_{\text{BOS}}, e_{x_\ell} \rangle = 1$, $\langle e_{x_\ell}, e_{x_m} \rangle = 1$ if $x_m = x_\ell$ and 0 otherwise. Due to the softmax, the mixing coefficient is $a = e/((k_{x_\ell} + 1)e + (L - k_{x_\ell}))$ (where $e$ is Euler's number) for comparing $x_\ell$ to $t_{\text{BOS}}$ and to all the tokens where $x_\ell = x_m$, and $b = 1/((k_{x_\ell} + 1)e + (L - k_{x_\ell}))$ otherwise, where, $k_{x_\ell} = hist_{\mathbf{x}}(\ell)$. Hence, the mixed token is:

$$\bar{x}'_\ell = ae_{\text{BOS}} + ak_{x_\ell}\bar{x}_\ell + \sum_{x_m \neq x_\ell} b\bar{x}_m + \bar{x}_\ell. \tag{20}$$

Applying $W_1$ and $b_1$, we obtain:

$$\begin{aligned} \gamma_\ell &= aT + ak_{x_\ell} + b(L - k_{x_\ell}) \\ &= aT + ak_{x_\ell} + 1 - a(k_{x_\ell} + 1) \\ &= a(T-1) + 1 \end{aligned} \tag{21}$$

since $(k_{x_\ell} + 1)a + (L - k_{x_\ell})b = 1$ by normalization via the softmax function. The value of $\gamma_\ell$ has a dependence on $k_\ell$ through $a$ and can be readout into the correct classification as shown Fig. 7. □

### A.2.3  (BOS; $p = 1$)

*Proof of Proposition 1 - bos.* We set $T = d > 2$ with $L \geq 2$ and $p = 1$. The construction of the embeddings and $e_{\text{BOS}}$ is analogous to the construction from Section A.2.2 for bos+sftm in the same setting. However, since no softmax is applied, the mixing coefficients as outputs of $\mathbf{A}_{\text{dot+sftm}}$ for comparing $(x_\ell, t_{\text{BOS}})$ or $(x_\ell, x_m)$ where $x_\ell = x_m$ is $a = 1$. For $x_\ell \neq x_m$ it is $b = 0$. Then from inserting these values in equation 20 and applying $W_1 = e_{\text{BOS}}$ and $b_1 = -T$ we obtain

$$\gamma_\ell = k_{x_\ell}. \tag{22}$$

This clearly allows again the single neuron to be read off to the correct result similar to the construction from 8. □

Note that there is a simple alternative construction that uses the tagged embeddings from the constructive proof of Prop. 2.

*Alternative Proof of Proposition 1 - bos.* We set $T = d > 2$ with $L \geq 2$ and $p = 1$. We note that by setting $t_{\text{BOS}}$ to zero we can achieve equivalence to the model dot. Since according to Prop. 2 there exists a weight configuration for dot which solves the histogram task, this configuration will also solve the histogram task for bos with $t_{\text{BOS}} = 0$. □

### A.3  INVENTORY-BASED COUNTING

### A.3.1  (LIN; $p = T$).

*Proof of Proposition 3 - lin.* Assume that $T = d > 2$ with $L > 2$ and $p = T$ and the goal is to find a weight configuration for the model lin. As embeddings we directly use the orthonormal basis with $T$ vectors $e_t$ in $\mathbb{R}^d$, where vectors are the embeddings are for the $T$ tokens. We set

$$\mathbf{A}_{\text{lin}} = \begin{bmatrix} a & a & \cdots & a \\ a & a & & \\ \vdots & & \ddots & \\ a & & & a \end{bmatrix}; \quad W_1 = E; \quad b_1 = -1, \tag{23}$$

where $a = 1/L$. We start by writing $z_{\ell,t}$ for $t \in \{1, ..., p = T\}$, the $t$-th activation of the first hidden layer of the feed-forward module

$$z_{\ell,t} = \sum_{m=1}^{L} a_{\ell m} \langle e_{x_m}, e_t \rangle + \langle e_{x_\ell}, e_t \rangle - 1. \tag{24}$$

If $e_t = e_{x_\ell}$, we have

$$z_{\ell,t} = k_{x_\ell} a + 1 - 1 = a k_{x_\ell}, \tag{25}$$

where, $k_{x_\ell} = hist_{\mathbf{x}}(\ell)$, applying the ReLU to this scalar keeps its value unchanged. If $e_t \neq e_{x_\ell}$, we have

$$z_{\ell,t} = a k_{e_t} + 0 - 1 = a k_{e_t} - 1 \leq 0. \tag{26}$$

The right hand side of the above equation is negative given our choice of $a$, hence applying the ReLU returns 0. This means that, for each token in the input sequence, the contributions of orthogonal tokens cancel, leaving us with a single hidden hidden neuron activated. Hence the count can be read off from $\gamma_\ell$. Since only one neuron is activated at a time, the readout from the same procedure as in bos+sftm can be applied to all hidden neurons $z_{\ell,t}$ simultaneously, instead of only one. This allows the model to solve the histogram task. □

### A.3.2  (LIN+SFTM: $p = T$)

*Proof of Proposition 3 - lin+sftm.* Assume that $T = d > 2$ with $L > 2$ and $p = T$. With the statement already proven for lin, we note that we can construct $\mathbf{A}_{\text{lin+sftm}}$ such that it is equivalent to $\mathbf{A}_{\text{lin}}$ from equation 23 via

$$\mathbf{A}_{\text{lin+sftm}} = \begin{bmatrix} a & a & \cdots & a \\ a & a & & \\ \vdots & & \ddots & \\ a & & & a \end{bmatrix} = \text{softmax} \left( \begin{bmatrix} \alpha & \alpha & \cdots & \alpha \\ \alpha & \alpha & & \\ \vdots & & \ddots & \\ \alpha & & & \alpha \end{bmatrix} \right), \tag{27}$$

where $a = 1/L$ which implicitly defines a choice of $\alpha$. This means that the construction is equivalent to `lin` and it follows automatically that also `lin+sftm` can solve the histogram task. $\square$

### A.3.3 (DOT+SFTM: $p = d = T$)

**Proposition 6** (IC for `dot+sftm`). *For `dot+sftm` and given $L, T > 2$ there exists a configuration of weights which solves the histogram task for $p \geq T$ and $d \geq T$.*

*Proof for Proposition 6.* We assume $L, T > 2$ and $p = T$ and $d = T$ and we consider `dot+sftm`. As previously for `dot` in Prop. 2, we set the key and query matrix to the scaled identity $W_K = W_Q = d^{1/4} I_d$. We use an orthonormal basis of $\mathbb{R}^d$ to define the parameters $e_t \in \mathbb{R}^d$ for the the token embeddings. In $\mathbf{A}_{\text{dot+sftm}}$ the pre-softmax mixing weights will be 1 for equal and 0 for different tokens due to the unit-norm token embeddings. Defining $k_{x_\ell} = hist_{\mathbf{x}}(\ell)$ for brevity, after the softmax we have that

$$a_{lm} = \begin{cases} \frac{e}{(L - k_{x_\ell}) + e k_{x_\ell}} & x_m = x_\ell, \\ \frac{1}{(L - k_{x_\ell}) + e k_{x_\ell}} & \text{else.} \end{cases} \tag{28}$$

Hence, for $e_t \neq e_{x_\ell}$

$$\langle \bar{x}'_\ell, e_t \rangle = \frac{k_{e_t}}{(L - k_{x_\ell}) + e k_{x_\ell}} < 1, \tag{29}$$

while for $e_t = e_{x_\ell}$

$$\langle \bar{x}'_\ell, e_{x_\ell} \rangle = \frac{k_{x_\ell} e}{(k_{x_\ell} e + (L - k_{x_\ell}))} + 1, \tag{30}$$

where the extra summand comes from the residual connection. Hence, by setting

$$W_1 = E; \quad b_1 = -1, \tag{31}$$

and applying the ReLU activation, equation 29 will be 0, while equation 30 will implicitly give us the counts as:

$$k_{x_\ell} = (L\gamma_\ell)/(-e\gamma_\ell + \gamma_\ell + e) \tag{32}$$

While the final layer cannot immediately implement non-linear functions in $\gamma_\ell$, it can take advantage of the fact that $\gamma_\ell$ can take only $L$ different values, similar to how we constructed $W_2$ and $b_2$ in Section A.2.1. Since eventually we need to map the $L$ values of $\gamma_\ell$ to the counts $[1, \cdots, L]$ the linear output layer is sufficient to implement this non-linear discrete map. Fig. 7 shows an example for this map for a given example. This allows the model to solve the histogram task.

The statement for $p > T$ and $d > T$ follows as we can simply set the surplus of parameters in the hidden layer/embeddings to zero. $\square$

### A.4 MAPPING A SCALAR TO A CATEGORICAL ONE-HOT ENCODING

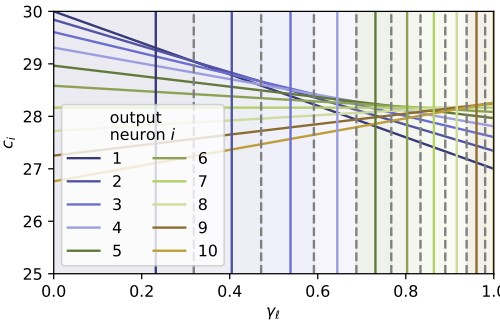

Figure 7: Demonstration of a hidden neuron output $\gamma_\ell$ which is mapped to $L = 10$ different neurons $c_i$ using a single output layer, according to the decision boundaries shown by the dotted lines. After applying the argmax function to the 10 output neurons, the highest value gives the discrete output. The solid lines mark the values a single hidden neuron would achieve for different counts in the explicit construction of the `dot+sftm` model.

It is straightforward to map a single scalar $\gamma_\ell$ to a series of neurons which activate one after another. This is needed as the second part of the feed-forward parameters to transform the count measured by the sum of the hidden neurons $\gamma_\ell$ to the discrete categorical representation of the output vector. Every output logit is a linear function of the hidden neuron's value. Since in our constructions we only map functions, where the ground truth output logit corresponds to an interval $[a, b] \in R$, the superposition of linear functions with increasing slope allows us to realize such a mapping. A visual sample is given in Fig. 7 for `dot+sftm`. In Fig. 8 we show the outputs for the `lin+sftm` model with the best accuracy for $T = 32$ for every $p, d$ ran in Fig. 1. While it is possible to learn the count from one hidden neuron only using inventory-based counting for each neuron, for some examples the count information seems to be spread out over several hidden neurons: The output logits are non-linear in the count and can hence not rely on a single hidden neuron only.

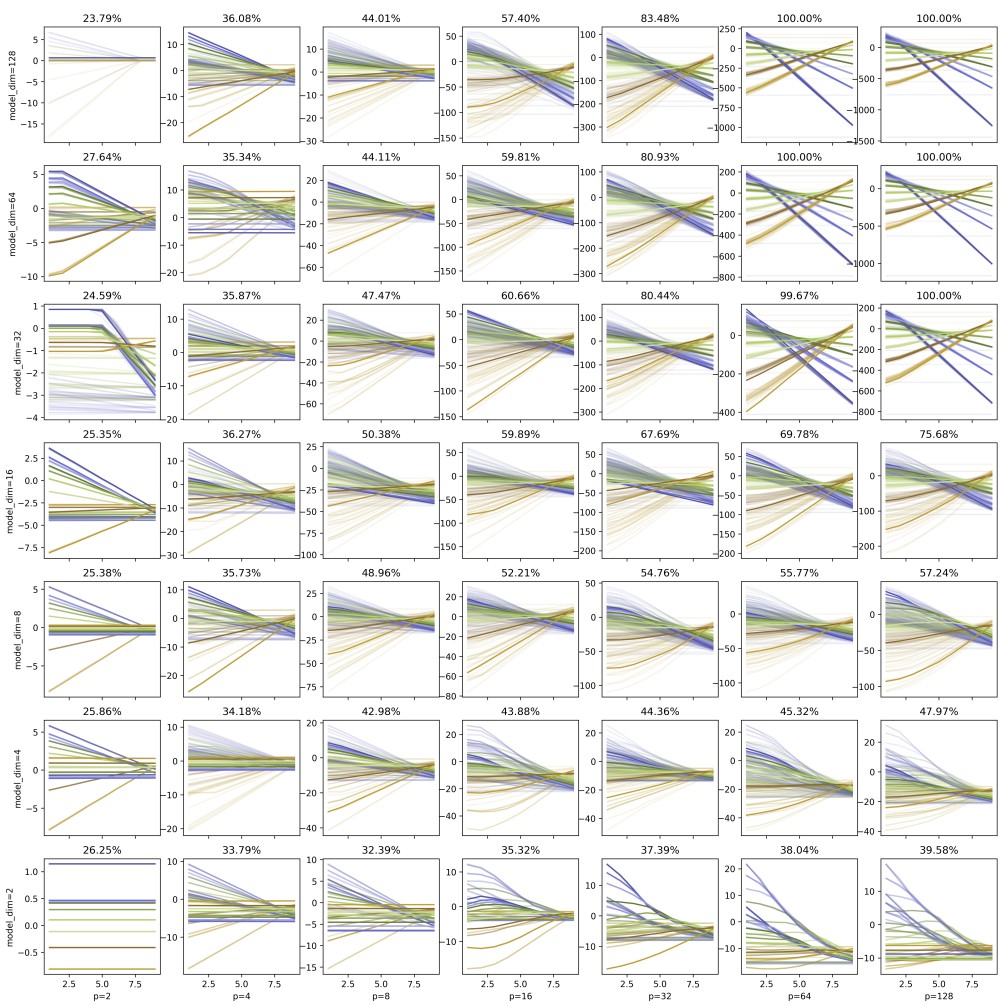

Figure 8: The output neurons $c_i(\mathbf{x}_\ell)$ visualized for examples of a learned version of `lin+sftm` for several model dimensions $d$ and hidden layer sizes $p$. Note that differently from Fig. 7, in this case the x-axis shows the number of occurrences $hist_\mathbf{x}(\ell = 0) = 1, \ldots, L - 1$ of the token $t$ in an input sequence $\mathbf{x} = [t, \cdots, t, v, \cdots, v]$ that contains otherwise only a token $v \neq t$ (and *not* the activation of a hidden neuron). We show the activations $c_i$ of the final layer output neurons activations (logits) in terms of the number of occurrences of a given token in the input. The colors represent the different output predictions and are as in the explicit construction from Fig. 7. We show several activations for different tokens $t \in \mathcal{T}$, where $T = 32$, and we highlight one of the example tokens $t$ with a wider line. While similar to the explicit construction from Fig. 7, the models with 100% accuracy are not necessarily linear in the count.

# B  EXPLICIT CONSTRUCTIONS FOR LINEARLY DEPENDENT EMBEDDINGS $d < T$

## B.1  OVERVIEW

In this section, we discuss the scenario when $d < T$, i.e. when the embeddings are necessarily linearly dependent. In that case, we can no longer assume that there exist embeddings with $\langle e_t, e_s \rangle = 0$ for all $t \neq s$. Nonetheless, also in this regime for some models it is possible to provide explicit constructions of the weights that have 100% accuracy. This relies on the fact that the prediction problem is inherently discrete, i.e. it chooses exactly one among $L$ classes. When we examine $\gamma_\ell \in \mathbb{R}$ from equation 6 which is mapped to the discrete class through the readout layer (see for example Fig. 7), we notice that the class boundary (the gray dashed class borders) can be placed variably in the margin between the values that $\gamma_\ell$ assumes for different counts $k_{x_\ell}$ (solid lines). In the following explicit constructions, our goal is to design embeddings with $d < T$ in such a way that we maximize the aforementioned margin: there will be pairs of token embeddings in the alphabet that have non-zero similarity $\langle e_t, e_s \rangle$, and in equation 6 this will create non-zero terms that will alter the value of $\gamma_\ell$. This means that for every possible sequence with $k$ occurrences of token $x_\ell$, the hidden activation $\gamma_\ell$ will assume values in a certain range $[\gamma_\ell^{\text{lower}}(k), \gamma_\ell^{\text{upper}}(k)]$. If these ranges overlap for different $k$, the count cannot be identified. However, we construct embeddings such that every for every $k = 1, \ldots, L-1$ it holds that

$$\gamma_\ell^{\text{upper}}(k) < \gamma_\ell^{\text{lower}}(k+1), \tag{33}$$

so we can still use a construction as in Fig. 7 to correctly compute the final count. In the remainder of this section, we introduce explicit constructions with $d < T$ for a given $L$, both for the cases where we have relation-based counting and inventory-based counting (the same argument as above transfers to $z_{\ell,t}$ from equation 24). Notably, for the explicit constructions we propose, the function of the lowest achievable $d(p, T, L)$ differs across different mixing types. To summarize:

- For models with **A** constant in the inputs or models *without* softmax activation, our explicit construction relies on an embedding matrix with a small mutual coherence. The mutual coherence is a concept from compressed sensing and coding theory that ensures that the maximal similarity between pairs of vectors is small (Donoho & Elad, 2003). We can upper bound the mutual coherence that the margins of the construction can tolerate to still achieve perfect accuracy in terms of a given $L$. At the same time, the mutual coherence of a set of vectors is naturally lower bounded in terms of the number of vectors $T$ and their respective dimension $d$, known as the Welch bound (Welch, 1974). When this bound can be attained and $T, L$ are given, this leads to the following bounds on $d$ for the different models, as outlined in Prop. 4, as

  $(\texttt{lin}, \texttt{lin+sftm}; p = T)$: $\left\lceil \frac{T(2L-3)^2}{T-1+(2L-3)^2} \right\rceil \leq d$,

  $(\texttt{dot}, \texttt{bos}; p = 1)$: $\left\lceil \frac{T(2L-3)^2}{T-1+(2L-3)^2} \right\rceil + 1 \leq d$,

  $(\texttt{dot}, \texttt{bos}; p = T)$: $\left\lceil \frac{T(L-1)}{T-1+(L-1)} \right\rceil \leq d$.

- For $\texttt{bos+sftm}$ we rely on the fact that the softmax function accentuates the largest value and thereby can drive attention scores for equal tokens $a_{ii}$ higher relative to attention scores of non-equal tokens $a_{ij}$. This distinguishes it from the previous case, and allows us to state Prop. 5 for which we describe an explicit construction that solves the histogram task with

  $(\texttt{bos+sftm}; p = 1)$: $d \geq \lceil \log_2(T+1) \rceil + 2$.
  $(\texttt{dot+sftm}; p = T)$: $d \geq \lceil \log_2(T+1) \rceil + 2$.

  Notably there is no explicit dependence on $L$ for the dimension. However, the smaller the dimension $d$ the more accurate computations and softmax numerical stability are required, as the softmax temperature depends on $L$. With infinitely precise computations we show it is even possible to achieve perfect accuracy with $d = 4$, but for finite computations this might pose a problem when $L$ becomes too large.

### B.2 EXPLICIT CONSTRUCTION FOR BOUNDED MUTUAL COHERENCE

We define the *mutual coherence* $\mathcal{M}$ of a set of $T$ unit norm vectors $\{v_1, \ldots, v_T\} \subset \mathbb{R}^d$ as

$$\mathcal{M} = \max_{i \neq j} |\langle v_i, v_j \rangle|. \tag{34}$$

This value is lower bounded for a given matrix by the Welch bound (Welch, 1974)

$$\mathcal{M} \geq \sqrt{\frac{T - d}{d(T - 1)}} = \mathcal{W}(T, d), \tag{35}$$

and equality can only be attained if $T < d^2$ (Strohmer & Heath, 2003). There is a large body of work in coding theory and compressed sensing concerning the existence and construction of a set of vectors that attains $\mathcal{M}$ at or close to $\mathcal{W}(T, d)$. Explicit constructions exist but are not known for every combination of $T$ and $d$. A list with existing constructions for the real space for small $T, d$ can be found in Fickus & Mixon (2016), but otherwise gradient-based optimization has been used to find good candidate matrices (Jiang et al., 2017; Jyothi & Babu, 2022).

In order to prove Prop. 4, we use the following idea: For a given $T$, $L$ and $p$, we can derive an upper bound on the mutual information of the embeddings in terms of $L$, which is required to obtain perfect accuracy. The form of this upper bound depends on the precise mixing strategy and the choice of $p$. Through the Welch lower bound on $\mathcal{M}$ we can in turn obtain a lower bound on $d$ in terms of $L$ and $T$. Note that the Welch bound cannot be attained for $T < d^2$ and in this case the bound on $d$ is strict.

#### B.2.1 (LIN, LIN+SFTM; $p = T$)

*Proof of Proposition 4 - lin.* To show the bound on $d$, we analyze the inventory-based construction for lin in equation 23. Given that $p = T$, and $L > 2$ is given, let us assume that there exists set of $T$ unit norm vectors $\{e_1, \ldots, e_T\} \subset \mathbb{R}^d$ with mutual coherence $\mathcal{M}$. We use these vectors as our embeddings.

The value $z_{\ell, t}$ for $t = x_\ell$, with $W_1 = [e_1, \ldots, e_T]$ and $b_1 = -1$ is

$$z_{\ell, t} = ak_{x_\ell} + a \sum_{m : x_m \neq t} \langle e_{x_m}, e_t \rangle, \tag{36}$$

and using that fact that the mutual coherence bounds the absolute value of the inner product

$$ak_{x_\ell} - a\mathcal{M}(L - k_{x_\ell}) \leq z_{\ell, t} \leq ak_{x_\ell} + a\mathcal{M}(L - k_{x_\ell}). \tag{37}$$

Similarly, for $t \neq x_\ell$ and $a = 1/L$ it still holds that

$$z_{\ell, t} \leq ak_t + a(L - k_t)\mathcal{M} - 1 + \mathcal{M} \leq 0, \tag{38}$$

provided that $\mathcal{M} < 1/(L + 1)$, for the worst case where $k_t = L - 1$. This means that the ReLU sets all hidden neurons $z_{\ell, t}$ to zero when $t \neq x_\ell$, and are therefore no contribution to the final result. Then, defining

$$\gamma_\ell^{\text{lower}}(k) = ak - a\mathcal{M}(L - k), \tag{39}$$
$$\gamma_\ell^{\text{upper}}(k) = ak + a\mathcal{M}(L - k), \tag{40}$$

we have that indeed for a sequence where $x_\ell$ occurs $k = 1, \ldots, L - 1$ times it holds that

$$0 \leq \gamma_\ell^{\text{lower}}(k) \leq \gamma_\ell(k) \leq \gamma_\ell^{\text{upper}}(k). \tag{41}$$

The first inequality is required due to the ReLU and holds when $\mathcal{M} < 1/(L - 1)$. From equation 33 we have the condition that for all $k = 1, \ldots, L$ it holds that

$$\gamma_\ell^{\text{upper}}(k) < \gamma_\ell^{\text{lower}}(k + 1), \tag{42}$$
$$k + (L - k)\mathcal{M} < (k + 1) + (L - k - 1)(-\mathcal{M}), \tag{43}$$
$$\mathcal{M} < \frac{1}{2(L - k) - 1}, \tag{44}$$

and since we assume that there exist at least two different tokens in the sequence, minimizing the bound over $k$ leaves for $k = 1$

$$\mathcal{M} < \frac{1}{2L - 3}. \tag{45}$$

which is valid provided that $L \geq 2$. Collecting all previous bounds on $\mathcal{M}$, we conclude that when $L \geq 4$ the above construction achieves the correct counts with $\mathcal{M} < \frac{1}{2L-3}$.

The Welch bound equation 35 gives an upper bound on $\mathcal{M}$ in terms of $T, d$ and therefore yields the final condition

$$d \geq \left\lceil \frac{T(2L - 3)^2}{T - 1 + (2L - 3)^2} \right\rceil \tag{46}$$

under which the given weight configuration is able to solve the histogram task with perfect accuracy.
$\square$

For `lin+sftm` the construction and conditions transfer directly, when the constant $\mathbf{A}_{\text{lin+sftm}}$ is constructed to match $\mathbf{A}_{\text{lin}}$ exactly.

### B.2.2 (DOT, BOS; $p = 1$)

*Proof of Proposition 4 - dot, $p = 1$.* We assume that $L > 2$ and $T$ given and we use a similar idea as the relation-based weight configuration from the proof of Prop. 2 for dot with $p = 1$. For the token embeddings, we assume that we have a set of $T$ unit norm vectors with $\{v_1, \ldots, v_T\} \subset \mathbb{R}^{d-1}$ with mutual coherence $\mathcal{M}$, where $d > 2$. We set the entries of the $T$ embedding vectors $e_t \in \mathbb{R}^d$ to be

$$e_t = \begin{bmatrix} v_t \\ \alpha \end{bmatrix}. \tag{47}$$

The shared counting subspace is defined on the last coordinate of the vectors via $e_{cnt} = [0, 0, \ldots, 1/\alpha]$. Then

$$\langle e_t, e_t \rangle = 1 + \alpha^2 \tag{48}$$

$$|\langle e_t, e_s \rangle| \leq \mathcal{M} + \alpha^2 \tag{49}$$

The mixed token with the residual connection at position $\ell$ for a given input sequence is

$$\bar{x}'_\ell = \sum_{m=1}^{L} \langle e_{x_m}, e_{x_\ell}, \rangle e_{x_m} + e_{x_\ell} \tag{50}$$

and the single hidden neuron $\gamma_\ell$ for a bias term $b_1 = 0$ and $W_1 = e_{cnt}$

$$\gamma_\ell = \langle e_{cnt}, \bar{x}'_\ell \rangle = \sum_{m=1}^{L} \langle e_{x_m}, e_{x_\ell}, \rangle \langle e_{x_m}, e_{cnt} \rangle + \langle e_{x_\ell}, e_{cnt} \rangle \tag{51}$$

$$= k_{x_\ell}(1 + \alpha^2) + \sum_{m : x_m \neq x_\ell} \langle e_{x_m}, e_{x_\ell}, \rangle + 1 \tag{52}$$

So that we can achieve for a given count $k$ the $\gamma_\ell^{\text{lower}}(k) \leq \gamma_\ell(k) \leq \gamma_\ell^{\text{upper}}(k)$ with

$$0 \leq \gamma_\ell^{\text{lower}}(k) = k(1 + \alpha^2) - (L - k)(\mathcal{M} + \alpha^2) + 1, \tag{53}$$

$$\gamma_\ell^{\text{upper}}(k) = k(1 + \alpha^2) + (L - k)(\mathcal{M} + \alpha^2) + 1. \tag{54}$$

We achieve the upper bound from zero, when $\mathcal{M} < 2/(L - 1)$, assuming that $\alpha$ is close enough to zero so that is is negligible. Finally, the condition from equation 33 yields

$$\mathcal{M} < \frac{1}{2(L - k) - 1} - \frac{2(L - k) - 2}{2(L - k) - 1} \alpha^2 \tag{55}$$

under the condition that $0 < \alpha < \sqrt{\frac{1}{2(L-k)-2}}$. Again, assuming there exist at least two different tokens in the sequence, the r.h.s. of the above expression is minimized for $k = 1$ as

$$\mathcal{M} < \frac{1}{2L-3} - \frac{2L-4}{2L-3}\alpha^2 \tag{56}$$

which is always positive assuming $L \geq 2$. This is the relevant bound when we have a large enough $L \geq 5$ and again $\alpha$ is close enough to zero. Again, combining this with the Welch bound equation 35 leads to

$$d - 1 \geq \left\lceil \frac{T(\frac{2L-3}{1-(2L-4)\alpha^2})^2}{T - 1 + (\frac{2L-3}{1-(2L-4)\alpha^2})^2} \right\rceil , \tag{57}$$

and when we choose $\alpha > 0$ close to zero, as for `lin` before

$$d \geq \left\lceil \frac{T(2L-3)^2}{T - 1 + (2L-3)^2} \right\rceil + 1. \tag{58}$$

$\square$

This proof holds equivalently for `bos` when we set the BOS token embedding to zero.

### B.2.3  (DOT, BOS; $p = T$)

We can decrease the required dimension $d < T$ even further than previously, when we have $p = T$ and implement inventory-based counting in the `dot` model (and equivalently in the `bos` model). In that case, the lower bound on $d$ becomes more loose, because we combine the ideas we saw in `lin` and $p = T$ for inventory-based counting and the effects on the margin in `dot` and $p = 1$.

*Proof of Proposition 4 -* `dot`, $p = T$. In our construction, for a given $T$ and $L$, we assume that there is a set of $T$ unit norm vectors $\{e_1, \ldots, e_T\} \subset \mathbb{R}^d$ with mutual coherence $\mathcal{M}$ upon which we build our embeddings. Note that the only difference to the previous relation-based case $p = 1$ is that this time there is no extra counting direction. Importantly, we set $K = d^{1/4}I_d$ as before, but $Q = \frac{1}{L}d^{1/4}I_d$. This gives an extra factor in the attention scores. Further, we set $b_1 = -1$ and the columns of $W_1 \in \mathbb{R}^{d \times T}$ to the embeddings $e_t$, as we did for `lin`. This results in a mixed token $\bar{x}'_\ell$ according to equation 50. The hidden neuron is

$$z_{\ell,t} = \frac{1}{L}\sum_{m=1}^{L}\langle e_{x_m}, e_{x_\ell}\rangle\langle e_t, e_{x_m}\rangle + \langle e_t, e_{x_\ell}\rangle - 1 \tag{59}$$

then with $t = x_\ell$ we have

$$z_{\ell,t} = \frac{1}{L}\left(k_{x_\ell} + \sum_{m:x_m\neq t}\langle e_{x_m}, e_t\rangle^2\right). \tag{60}$$

Note that the square in equation 59 is what differs from the $z_{\ell,t}$ in equation 36. This is because the term $\langle e_{x_m}, e_t\rangle$ is once introduced through the dot-product attention and once through the dot-product via $W_1$. Conversely, with $t \neq x_\ell$ it becomes

$$z_{\ell,t} = \frac{1}{L}\left(k_t\langle e_t, e_{x_\ell}\rangle + \sum_{m:x_m\neq t}\langle e_{x_m}, e_{x_\ell}\rangle\langle e_{x_m}, e_t\rangle\right) + \langle e_t, e_{x_\ell}\rangle - 1 \tag{61}$$

$$\leq \frac{1}{L}\left(k_t\mathcal{M} + (L-k_t)\mathcal{M}\right) + \langle e_t, e_{x_\ell}\rangle - 1 \tag{62}$$

$$\leq 2\mathcal{M} - 1 \tag{63}$$

if we set $\mathcal{M} < 0.5$, which we need anyways for $L \geq 2$ by the stronger upper bound on $\mathcal{M}$ that we derive in the following, we finally have for $t \neq x_\ell$

$$z_{\ell,t} < 0. \tag{64}$$

Again, negative $z_{\ell,t}$ are set to zero via the ReLU, and the final outcome $\gamma_\ell = z_{\ell,t=x_\ell}$ depends only on a single hidden neuron equation 59. This eventually leads to

$$0 \leq \gamma_\ell^{\text{lower}}(k) = \frac{k}{L}\,, \tag{65}$$

$$\gamma_\ell^{\text{upper}}(k) = \frac{1}{L}\left(k + \mathcal{M}^2(L-k)\right), \tag{66}$$

and using the same concept as before, while minimizing over $k$ and applying the Welch bound, to the upper bound

$$\mathcal{M} < \sqrt{\frac{1}{L-1}}\,. \tag{67}$$

The final bound is more loose than it was for $p = 1$ as we only require

$$d \geq \left\lceil \frac{T(L-1)}{T-1+(L-1)} \right\rceil . \tag{68}$$

$\square$

### B.3 EXPLICIT CONSTRUCTION WITH BINARY REPRESENTATIONS AND SOFTMAX

In our final analysis we examine the key difference between the models `bos+sftm` and `bos` – the softmax activation. In order to show Prop. 4 we needed to construct embeddings with a low mutual coherence, because the term $\langle e_t, e_s \rangle$ introduced an error on the mixed token, when $t$ and $s$ were not equal. Now, with the softmax activation applied to the mixing coefficients, the model can use the non-linearity of this transform to its advantage to separate the relative error.

Recall the softmax function is

$$\text{sftm}(\mathbf{z})_i = \frac{e^{z_i}}{\sum_{j=1}^{n} e^{z_j}} \quad \text{for} \quad i = 1, 2, \dots, n\,, \tag{69}$$

and when we compute $\text{sftm}(\kappa \mathbf{z})_i$ we say it is a softmax with a inverse temperature $\kappa > 0$. When $\mathbf{z}$ of length $L$ contains only two different values, one with $k$ and the other with $L - k$ occurrences, then as $\kappa \to \infty$ the mass concentrates only on the larger value of the two, and sets the other to zero. We use this intuition to create token embeddings that fulfill for all $t, s = 1, \dots, T$ and $s \neq t$

$$\langle e_t, e_t \rangle = 1\,, \tag{70}$$
$$\langle e_t, e_s \rangle < 1 + \epsilon\,, \tag{71}$$

where $\epsilon > 0$.

The idea is that the softmax with a high enough inverse temperature sets the term for different tokens, $\langle e_t, e_s \rangle$, *close enough* to zero, essentially eliminating the noise. Note that equation 70 is a weaker condition on the set of token embeddings than for example the bound of the mutual coherence in terms of the sequence length $L$ `bos` with $p = 1$ in Section B.2.2. It allows us to obtain perfect accuracy with smaller $d$. In the following, we describe the construction of the matrix explicitly.

The supplementary code at https://github.com/to-be-deanonymized contains executable `pytorch` models that have the weight configurations that are used to prove Propositions 5 and the Remark for $d = 4$, which allows one to test the devised weight configurations for fixed $T, L, d$ in practice.

### B.3.1 (BOS+SFTM; $p = 1$)

*Proof of Proposition 5 - `bos+sftm`.* For a given $T, L > 2$ we set the embeddings vectors to the binary representation of the token index $t = 1, \dots, T$ in $d' = \lceil \log_2(T+1) \rceil$ dimensions

$$e_t = \begin{bmatrix} \text{bin}(t)\langle \text{bin}(t), \text{bin}(t) \rangle^{-1} \\ \\ \alpha \\ 0 \end{bmatrix} \quad ; \quad e_{BOS} = \begin{bmatrix} \text{bin}(0)\langle \text{bin}(0), \text{bin}(0) \rangle^{-1} \\ \\ 1/\alpha \\ 1 \end{bmatrix}. \tag{72}$$

where $\mathrm{bin}(t) = [v_1, \ldots, v_{d'}] \in \{0, 1\}^{d'}$ with $t = \sum_{i=1}^{d'} v_i 2^{i-1}$. We select $\alpha > 0$. Then we have that

$$\langle e_t, e_t \rangle = 1 + \alpha^2 \,, \tag{73}$$

$$\alpha^2 \leq \langle e_t, e_s \rangle \leq \sqrt{1 - \frac{1}{d'}} + \alpha^2 \leq 1 + \alpha^2 - \epsilon \,, \tag{74}$$

$$\langle e_t, e_{BOS} \rangle = 1 \,, \tag{75}$$

where $\sqrt{\frac{d'-1}{d'}} = \langle e_{2^{d'}-1}, e_{2^{d'}-2} \rangle$, which has the largest overlap among all possible non-equal pairs of tokens, and the lower bound comes from all coordinates being positive. Using a readout on the direction only present in the $e_{BOS}$ token, namely, $W_1 = [e_{cnt}] = [0, \ldots, 0, 1] \in \mathbb{R}^d$ and $b_1 = 0$, we construct

$$\gamma_\ell = \langle e_{cnt}, \bar{x}'_\ell \rangle = \mathrm{sftm}(EE_\ell^T)_0 \langle e_{BOS}, e_{cnt} \rangle + \sum_{m=1}^{L} \mathrm{sftm}(EE_\ell^T)_{m+1} \langle e_{x_m}, e_{cnt} \rangle + \langle e_{x_\ell}, e_{cnt} \rangle \tag{76}$$

$$= \mathrm{sftm}(EE_\ell^T)_0 \tag{77}$$

$$= \mathrm{sftm}([\langle e_\ell, e_{BOS} \rangle, \langle e_\ell, e_1 \rangle, \ldots, \langle e_\ell, e_L \rangle])_0 \tag{78}$$

The goal of applying the softmax function is to diminish the contributions of error equation 74, while having the final dimension of the $e_{BOS}$ token be representative of the count of $x_\ell$. The maximum error is induced when the upper bound equation 74 is attained for all tokens in the sequence $\mathbf{x}$ that are not equal to $x_\ell$. The minimum error is obtained when these different tokens attain the lower bound. Without loss of generality on the ordering, this implies that for a given length $L$ and a softmax activation function with an inverse temperature $\kappa^2$, we have that

$$\gamma_\ell^{\mathrm{lower}}(k) = \frac{e^{\kappa 1}}{e^{\kappa 1} + k e^{\kappa(1+\alpha^2)} + (L-k) e^{\kappa(1+\alpha^2-\epsilon)}} \,, \tag{79}$$

$$\gamma_\ell^{\mathrm{upper}}(k) = \frac{e^{\kappa 1}}{e^{\kappa 1} + k e^{\kappa(1+\alpha^2)} + (L-k) e^{\kappa\alpha^2}} \,. \tag{80}$$

We explicitly need $\epsilon$ strictly greater than zero, since otherwise there is no information about the count in $\gamma_\ell$ when it becomes independent of the count $k$. Notice, that this time it holds that $\gamma_\ell$ that correspond to higher values correspond to smaller counts, since a larger count corresponds to a larger denominator, i.e. a smaller $\gamma_\ell$. Due to this inverse relationship, for this model, we want that for all counts $k = 1, \ldots, L-1$ that it holds that

$$\gamma_\ell^{\mathrm{upper}}(k+1) < \gamma_\ell^{\mathrm{lower}}(k) \,. \tag{81}$$

This can be achieved by setting the inverse temperature $\kappa$ accordingly.

In the following we show that there exists a $\kappa$ which fulfills equation 81 for all $d' \geq 2$ and $L > 2$. Observe that $\gamma_\ell^{\mathrm{upper}}(2) < \gamma_\ell^{\mathrm{lower}}(1)$ implies the bounds for all other $k$. We define the distance or margin as

$$\mathrm{dist}(\kappa) = \gamma_\ell^{\mathrm{lower}}(1) - \gamma_\ell^{\mathrm{upper}}(2) \,. \tag{82}$$

Since at $\kappa = 0$ both $\gamma_\ell^{\mathrm{upper}}(2) = \gamma_\ell^{\mathrm{lower}}(1) = 1/(L+1)$, the distance is zero. However then it becomes impossible to distinguish $k = 1$ and $k = 2$, as they receive the same weight. We therefore need the additional condition that $\gamma_\ell^{\mathrm{upper}}(2) \neq \gamma_\ell^{\mathrm{lower}}(1)$. At $\kappa = 0$, we observe that this function

---

[2]In order to introduce the inverse temperature $\kappa$ of the softmax in the model, we scale the query matrix. We set $K = d^{1/4} I_d$, but $Q = \kappa d^{1/4} I_d$.

has a negative derivative, as

$$\frac{\partial}{\partial \kappa}\text{dist}(\kappa)|_{\kappa=0} = \text{sftm}(\kappa z_{\text{lower}})_0 \left( (z_{\text{lower}})_0 - \sum_{i=0}^{L+1} (z_{\text{lower}})_j \text{sftm}(\kappa z_{\text{lower}})_i \right) \tag{83}$$

$$- \text{sftm}(\kappa z_{\text{upper}})_0 \left( (z_{\text{upper}})_0 - \sum_{i=0}^{L+1} (z_{\text{upper}})_j \text{sftm}(\kappa z_{\text{upper}})_i \right) \tag{84}$$

$$= -\left( \frac{1}{L+1} \right)^2 \left( \left[ (1+\alpha^2) + (L-1)(1+\alpha^2 - \epsilon) \right] - \left[ 2(1+\alpha^2) + (L-2)\alpha^2 \right] \right) \tag{85}$$

$$= -\left( \frac{1}{L+1} \right)^2 \left[ (L-2) - (L-1)\epsilon \right] \tag{86}$$

$$< 0 \tag{87}$$

where the last bound is met when $0 < \epsilon < 0.5$ which is fulfilled already for $d' = 2$ and when $L > 2$. As the distance function is continuous, there exists a $\kappa$ close to zero for which the $\text{dist}(\kappa) < 0$. Simultaneously, as $\kappa \to \infty$, we have that due to the concentration of the softmax probabilities on the largest entry, which here is $1 + \alpha^2$, it holds that as $\kappa \to \infty$ we have $\text{dist}(\kappa) \to 0$. At the same time, the function approaches infinity from the positive regime. For large enough $\kappa$ we have $\gamma_\ell^{\text{upper}}(2) < \gamma_\ell^{\text{upper}}(1)$.

When we select the smallest possible $\kappa > 0$, we avoid computing functions with large exponential terms. To find the non-trivial root of $\text{dist}(\kappa)$ numerically, we consider a simplification of equation 82. We define $u = e^\kappa$. Then it holds that we can solve

$$\text{dist}(\kappa) = 0 = (L-1)u^{(1-\epsilon)} - u - (L-2) \tag{88}$$

numerically for $\kappa > 0$. This shows that we can find an explicit construction with 100% accuracy with $p = 1$ and $d' > 2$ for the `bos+sftm` when we have

$$d = \lceil \log_2(T+1) \rceil + 2. \tag{89}$$

For example, for the case of $L = 10$ and $T = 32$ this allows for a dimension $d = 7$ with $\alpha = 0.01$ (and for $T = 31$ with the same settings $d = 6$ suffices). $\qquad\square$

**Remark** ($d = 4$). In principle, it is enough to have some $\epsilon > 0$ that ensures that overlaps between different token embeddings are strictly less than one. In principle, we can find an arbitrary number of tokens $T$ that satisfy this condition for just $d' = 2$. Take for example the following construction. For $t = 1, \ldots, T$ tokens with $T$ odd we can design the set of embeddings

$$v_t = \begin{bmatrix} \sqrt{\frac{t}{T}} \\ \sqrt{\frac{T-t}{T}} \end{bmatrix}. \tag{90}$$

Each $\langle e_t, e_t \rangle = 1$ and for $t \neq s$ the overlap $\langle e_t, e_s \rangle \leq \langle e_{(T+1)/2}, e_{(T-1)/2} \rangle = \sqrt{T^2 - 1}/T$.

This implies that $\epsilon \to 0$ as $T \to \infty$ at a rate $1/T$. Since smaller $\epsilon$ imply larger values of the temperature to solve equation 88, this might become problematic when this exceeds the accuracy of computations. Previously, for the binary representation construction from equation 72, we had that $\epsilon$ shrinks at a rate $\sim 1/\log_2(T)$. For the intermediate regime between $\log_T(T) + 1$ and $\log_2(T)$ dimensions, one can generalize this principle to arbitrary bases, e.g. $\log_3(T) > 2$, resulting in a smaller dimension but also less favorable (smaller) $\epsilon$ – this construction thus comes with a clear trade-off.

### B.3.2 (`DOT+SFTM`; $p = T$)

*Proof of Proposition 5 - `dot+sftm`.* For this model, the explicit construction is analogous to the previous one. Instead of using $p = 1$ we use $p = T$. The selection of the embeddings is analogous, but instead of a counting direction we read off all the weight directions separately with $T = d$. Not having a counting direction also saves the additional two dimensions required for `bos+sftm` with $p = 1$. In the feed-forward layer with $W_1$ the explicit construction considers again $z_{\ell,t}$ for every token $t \in \mathcal{T}$. The selection of the temperature is also analogous, with the exception that one has $L$ terms in the softmax instead of $L + 1$. $\qquad\square$

## C   DATA GENERATION

Every sample $\mathbf{x} = (x_1, \cdots, x_L)$ is generated recursively as follows, starting from size $K = L$ and alphabet $\mathcal{T}' = \mathcal{T}$:

1. Sample an integer $k$ uniformly from $[1, \cdots, K]$.

2. Sample a token $t$ uniformly from $\mathcal{T}'$.

3. Set $x_i = t$ for all $i = k, \cdots, K$.

4. Set $\mathcal{T}' = \mathcal{T}' \setminus \{t\}$ and $K = k$.

5. If $K \neq 0$, repeat from 1.

6. Set $\mathbf{x} = \text{shuffle}(\mathbf{x})$.

In contrast to sampling the elements of each sequence uniformly at random from the alphabet, this simple strategy enables us to better control the distribution of counts in the training dataset.

## D   ADDITIONAL EXPERIMENTS

### D.1   BEST ACCURACY

In Fig. 9, we show the best reached accuracy during training over the five sample runs. This gives insights into the feasibility of implementing a counting solution for a given combination of parameters $T, d, p$ of a model.

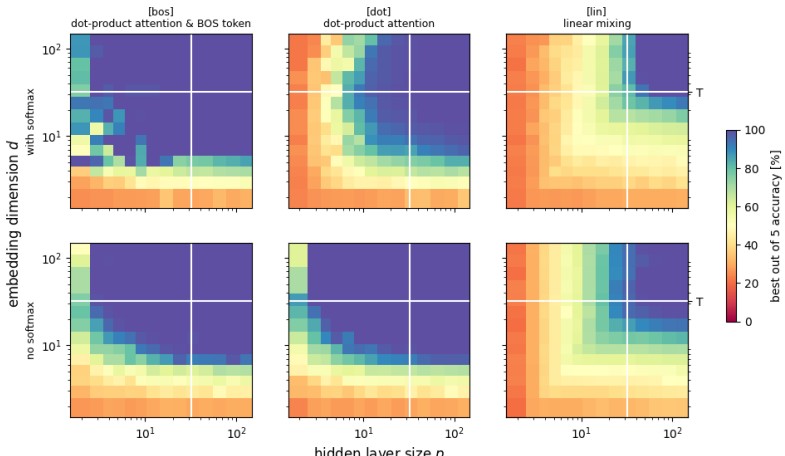

Figure 9: Experiments from Fig. 1 ($T = 32$), we show only the *best accuracy* during training reached from the 5 randomly initialized runs per model/hyperparameter configuration.

### D.2   VARIABILITY

In Fig. 10 we explore the influence of initialization on the performance via the variability of the final accuracy for several runs. Especially in the $p, d < T$ regime where `bos+sftm` is able to reach an accuracy relatively close to 100%, the variability of the accuracies resulting from different initializations is quite large.

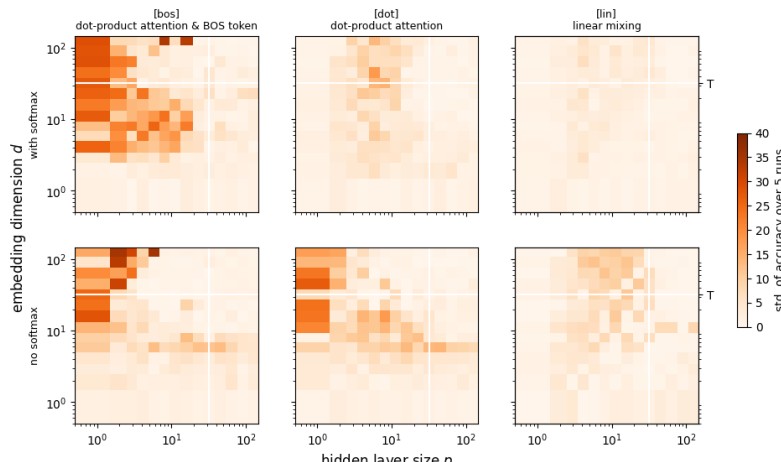

Figure 10: Experiments from Fig. 1 with $T = 32$, standard deviation of the accuracy reached after training from the 5 randomly initialized runs per model/hyperparameter configuration.

### D.3 MODEL WITH ALTERNATIVE $L = 15$

We repeat the experiments presented in Fig. 1 for $L = 15$ in Fig. 11, leading to the same phenomenology, in line with our hypothesis that indeed the number of tokens $T$ determines the relevant transition point, and not the sequence length $L$. However, the accuracy is comparatively worse when no high-accuracy solution is reached.

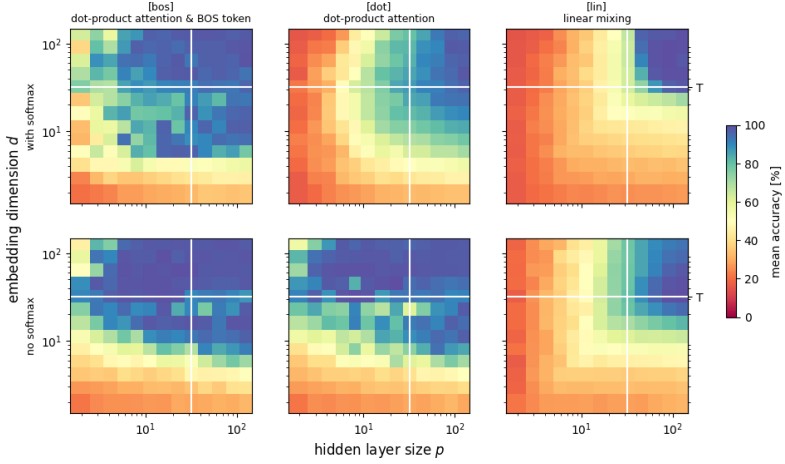

Figure 11: Experiments as in Fig. 1, but with sequence length fixed to $L = 15$.

### D.4 MODELS WITH TWO LAYERS

In this section, we look at the case where we have models that have an extra layer, i.e. instead of the logit output layer after the feed-forward part, we add another layer with the same dimensionality $d$ as the previous layer – the same mixing and the same hidden layer size – to then lead into the classification. Of course the parameters are not shared between the layers. Note that this model does not have an extra residual in the MLPs.

We train the model in the same setting as in the main and report the results for the different architectures, this time with 2 layers, in Fig. 12. To compare more easily with the previous set-up, we show the difference between the single and double layer case in Fig. 13. Remarkably, the general picture does not seem to change significantly. Indeed, the two layer model is generally better, extending the

range where perfect models can be found slightly, but the general trend remains. Given this coarse grained experiment we hypothesize, that the extra layer aids the optimization process, and improves robustness in the regions where and the softmax is used to disentangle non-orthogonal embeddings. More generally, these results are not as comprehensive as our previous results as they are note supported theoretically beyond a single layer. They warrant more detailed in further work with more layers and realistic settings.

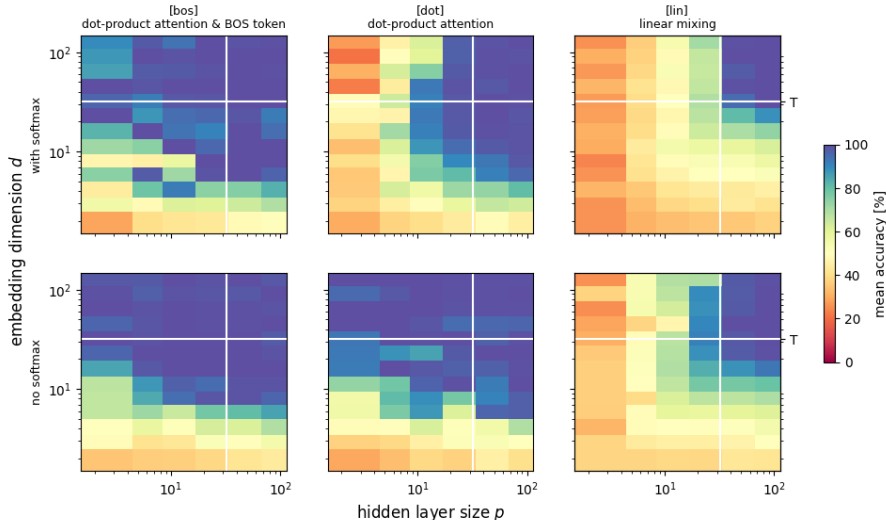

Figure 12: Experiments as in Fig. 1, but for fewer values of $p$ and $d$, as well as models where the layers are repeated as described in Section D.4.

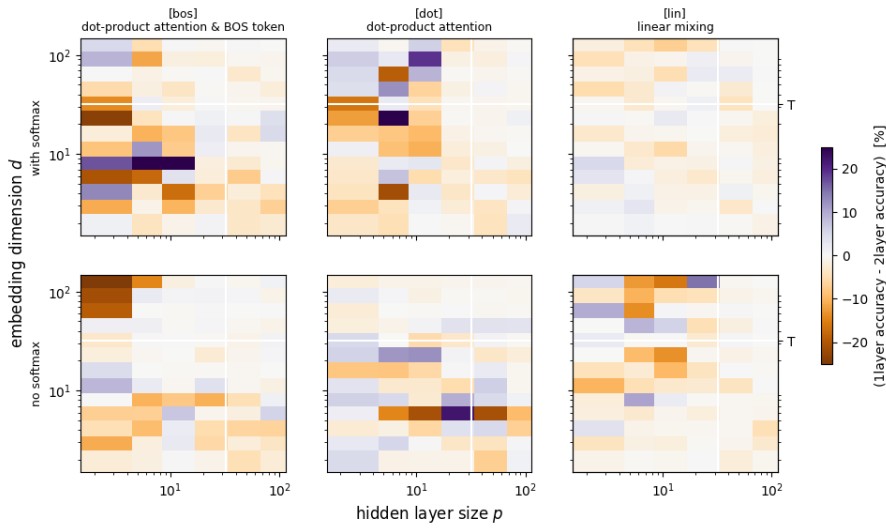

Figure 13: Difference between the accuracy of a single and two layer attention model, for different mixing layers and hyperparameter setups. Experiments as in Fig. 1 for a single layer attention model, and as in Fig. 12 for the two layer model.

## D.5 MODEL WITH RANDOM BUT FIXED EMBEDDINGS

In Fig. 14, we repeat the experiments of Fig. 1, but for embeddings that are frozen throughout training (also 5 runs). In the regime $d < T$ where there is no mutual orthogonality possible, the

random embeddings result in worse performance than the learned ones. Especially for `bos+sftm`, learning the embeddings increases the performance strongly in some regimes. This indicated that the models indeed learn adapted embeddings here.

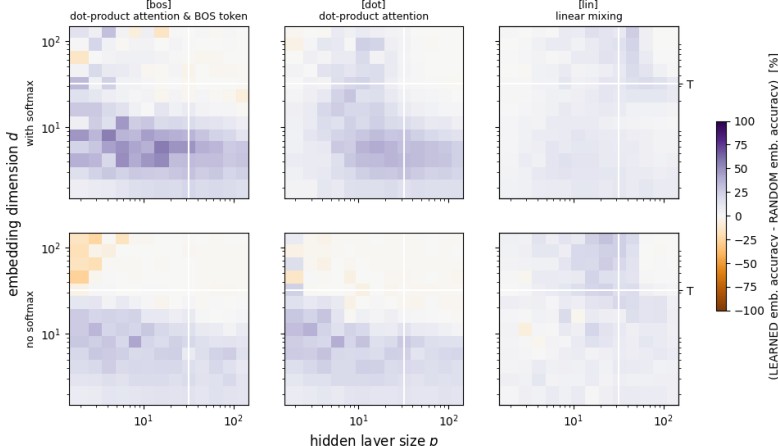

Figure 14: The difference between learned and random embeddings for $T = 32$. Orange indicates that the random embeddings perform better on average. Purple indicates that the learned embeddings perform better on average. Experimental settings as in Fig. 1.

## D.6 BOS MIXING TOKEN

In Fig. 3 in the main, we describe how the $t_{\mathrm{BOS}}$ is the main predictor for the count. Here, we provide more evidence by showing how the count predictions for mixed tokens $\bar{x}'$ output by the feature transform $f$ are invariant to the type of other token present in the mixed token. The results for four different tokens are shown in Fig. 15.

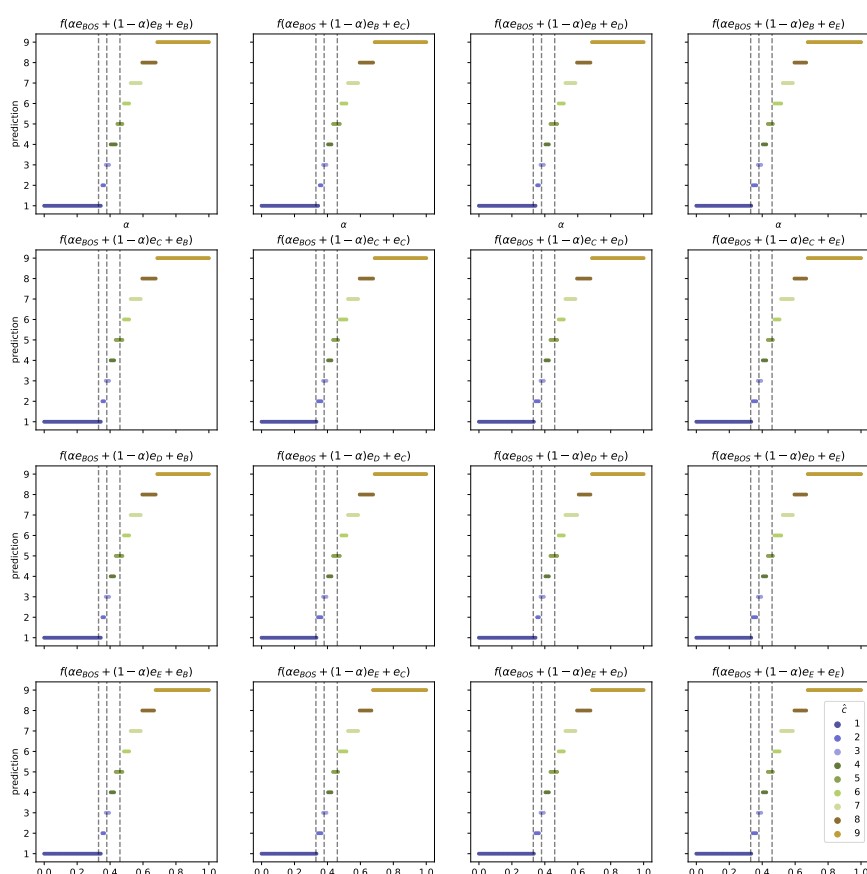

Figure 15: For the same model as in Fig. 3, we vary the inputs to the feature transformation $f$ to show it is independent on the precise input sequence, but only depends on the prevalence of $t_{\text{BOS}}$. We vary the inputs between the learned tokens $[B, C, D, E]$.

### D.7 SINGULAR VALUE DECOMPOSITION OF $W_1$

In Fig. 16 we show the distribution of singular values of $W_1$ for several runs of the model to investigate whether models that are capable of both IC and RC are implementing the more memory heavy IC or the same solution that they can find for $p = 1$ with RC.

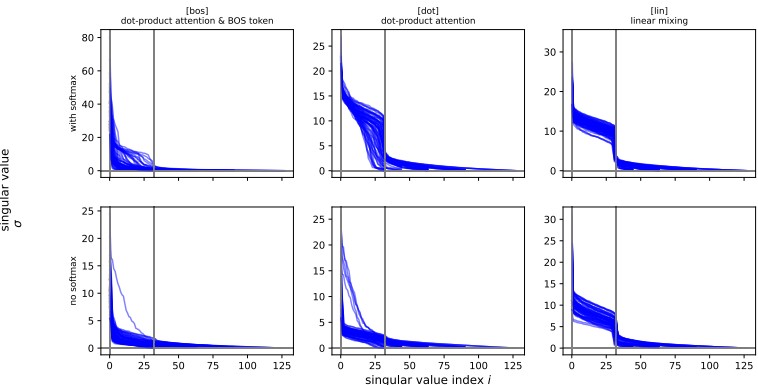

Figure 16: *Singular values of $W_1$.* We show the results for all models from Fig. 1 with $T = 32$, where $p, d \geq T$ and the accuracy is at least 99%. Some qualitative differences are visible for `bos` and `dot`.

