# OpenReview forum: "Counting in small transformers: The delicate interplay between attention and feed-forward layers"
_ICLR.cc/2025/Conference — Submitted to ICLR 2025_

### Official Review · Reviewer_aS5m · 2024-10-24

**Soundness:** 3
**Presentation:** 3
**Contribution:** 2
**Rating:** 6
**Confidence:** 4

**Summary:**

In an attempt to improve the interpretability of different elements of transformers, the authors use transformers to solve a histogram task. They incorporate two different counting strategies (relation-based and inventory-based), and replace different components of transformers in a controlled setting to study the relative role of each component. They find that both the attention mechanism (without applying softmax) and a sufficiently large FFN are capable of solving the histogram task. They also find that this task is sensitive to hyperparameters.

**Strengths:**

- The explanations, logic and experiments are sound

**Weaknesses:**

- If I understood correctly, the main contributions are that for the histogram test: 1. using the attention mechanism but without the softmax step (relation-based counting) is capable of high accuracy, and 2. Using learnable-matrix multiplication instead of attention (inventory-based counting) is not capable of high accuracy, but using a FFN can mitigate this by acting as a lookup table. It would be helpful if the authors would elaborate on why these findings are important.
- It would also be helpful if the authors provide some intuition on why they chose the histogram test. To be specific, transformers are designed for prediction and regression tasks. It's not clear why the authors chose to study a task that diverge from conventional transformer use cases. Perhaps there are interesting theoretical or practical insights that motivated this direction?

**Questions:**

- In your opinion, what do these findings tell us about the hypothesis space of transformers?

---

> ### Author Response · Authors · 2024-11-19
>
> Dear reviewer aS5m,
>
> We thank you for your review and feedback. We are glad to hear that you appreciate the experiments and explanations for our analysis of the histogram task.
>
> Regarding your concerns, the implications for practical insights are shared with other reviewers and we answer them in more detail in the second part of the common reply. We answer the other points you raised below:
>
> > **Relevance of the relation vs. inventory-based counting mechanisms**
>
> An important difference between the two mechanisms is that relation-based counting is much more parameter efficient than inventory-based counting. The full alphabet does not need to be saved as an inventory in the feed-forward MLP. In the real world, parameter efficient solutions are less compute costly and hence preferable. Beyond this, understanding the model internals can be useful in several ways. As an example, imagine adding new tokens to a pretrained model and training again. Knowledge on which mechanism is at play inside the model might help us understand which parameters of the network need to be changed, and which can be left frozen. This might explain why a low-rank approximation might fail to generalize for an inventory-based algorithm. Even though this example must be taken with a grain of salt, it serves as an example to demonstrate how insights into the concrete mechanisms might ultimately aid our understanding.
>
> > **Underlying reasons to choose histogram task rather than prediction and regression**
>
> As this concern is also shared with some of the other reviewers, we answer this question more generally in the global reply. Let us just highlight here that  the histogram task itself is still a sequence prediction task, and that our main difference from real transformers comes from modeling the task not in an autoregressive manner. We think that this simplifying modeling choice is not too limiting in comparison to real-world models. Practically, in a summarization task, a transformer typically has full access to the information that should be aggregated through its full history of tokens. Indeed, for the counting task we give every token access to all items in the sequence that it is supposed to count, i.e. summarize. From a theoretical perspective, it is easier to manually construct weights that solve the task when we can disregard the position of the sentence. Of course, incorporating positional information would indeed bring us closer to real-world scenarios, but it would also make the resulting solutions even harder to interpret, which is a trade-off we wanted to avoid for the scope of this study.
>
> > **In your opinion, what do these findings tell us about the hypothesis space of transformers?**
>
> Our findings underscore the intricate interdependencies between the structure of a transformer block, the algorithms it can implement, and its efficiency in doing so. For our particular task and models we now have a good understanding of this interaction and we are for example able to predict how many parameters are needed to find a solution in different scenarios. Interestingly, our experiments also show that understanding the hypothesis space is not enough - we showed that in one case very few parameters suffice for a solution but the learning algorithm does not find it, showing that even when the hypothesis space is large enough, one might not find it.
>
> If you have **further comments**, questions, or would like to explore additional aspects of our work, we would be more than happy to engage in further discussion. Similarly, if our responses have adequately addressed your concerns, we kindly ask you to consider raising your score.

---

> > ### Comment · Reviewer_aS5m · 2024-11-23
> >
> > Thank you for your response. However, I'm still unsure about the significance of the results and will maintain my previous score.

---

### Official Review · Reviewer_yFeG · 2024-11-04

**Soundness:** 3
**Presentation:** 3
**Contribution:** 2
**Rating:** 6
**Confidence:** 3

**Summary:**

This work investigates how different components of transformer architectures contribute to solving the counting task (creating a histogram of token occurrences in a sequence). The key research question is: how do architectural choices like embedding dimension, hidden layer size, and mixing mechanism (dot product attention vs linear) affect the solutions a transformer can learn? The authors discover and theoretically back up two distinct strategies transformers can use: (1) relation-based counting (which is more efficient but requires specific architectural features) and (2) inventory-based counting (which requires more parameters but is more flexible) - and show how seemingly minor architectural choices like including softmax in attention can dramatically affect which strategy emerges. They map out exactly what parameter regimes allow each strategy to work, including surprising findings about how transformers can still function even when the embedding dimension is smaller than the vocabulary size.

**Strengths:**

* The paper is well-written, easy to follow, and provides sufficient analysis for its claims.
* Discovers and Discusses two strategies which 1-layer transformers can use to perform the counting task (i.e., relation-based and inventory-based) and in doing so considers a range of possible hyperparameter selections that affect this strategic choice.
* During its analysis further studies the role of BoS token showing that together with softmax in the counting task, can help models with smaller p and d sizes to still reach perfect accuracy.

**Weaknesses:**

* Limited scope and applicability, specifically: (1) Only examines single-layer transformers, (2)Focuses on just one task (histogram counting)
* No clear path to extending insights to more complicated settings more specifically multi-layer transformers
* Lack of practical impact in the sense that it is hard to find a direct translation from this paper findings to real-world transformer design for real-world tasks

**Questions:**

Have you done any analysis or insight you can share on:
1. what behaviour can we expect in terms of the same hyperparameter choices (embedding dimension, Hidden Layer Size, and Vocab Size) whether as a  2 or 3 layer blackbox and/or an isolation of one of the transformer layers in a multi-layer setting?
2. Have you done any initial analysis to share about whether the same relationships for parameters and potential solving strategies hold for other tasks (such as the sorting or lookup problems you referred to or any other tasks)

---

> ### Author Response · Authors · 2024-11-19
>
> Dear reviewer yFeG,
>
> We thank you for your appreciation for our results on the relation- and inventory-based mechanisms as well as the impact of entangled embeddings. We are glad our effort to clearly guide you through the rather subtle complexities of this histogram task was fruitful.
>
> **Limited scope of the histogram task:** Your concerns about the limited scope and the lack of practical insights are partially shared with other reviewers, and we answer them together in a more detailed common reply above.
>
> Let us just emphasize here that, for us, the in-depth analysis of this toy example reveals how common intuitions about transformers can be extremely misleading. For example, we show that while the sequence-to-sequence language RASP [https://arxiv.org/abs/2106.06981](https://arxiv.org/abs/2106.06981) is a great tool, it is not always sufficient to describe the capabilities of transformers. The softmax can be both detrimental and beneficial and the inductive bias of dot-product attention may be insufficient to save parameters when paired with the wrong architectural component. With the histogram task we present a well-understood cautious tale of how relying on our intuitions does not generalize to transformers.
>
> From this toy-starting point, extending these insights in future work requires taking a step closer to the larger multi-layer models you mention. Since the histogram task consists of steps that select, aggregate and summarize information from the context, it seems plausible that we can test the newly gained intuitions from our toy example in a setting with larger models for tasks with similar processes e.g. text summarization *(existence of inventories in MLPs? Robustness to slight non-orthogonality/perturbations? Role of aggregation variables like BOS? Tags for functions on learned embeddings?)*.
>
> Regarding your **Questions 1 and 2**, they lead us to run some extra experiments, with some new insights, so thanks for that!
>
> > What can we expect for similar choices of the hyperparameters for 2 or 3 layers, in a black box setting or/and the properties of individual layers in multilayer setups?
>
> Thanks to your suggestion we ran preliminary experiments on the histogram task again for length $L=10$ and $T=32$ tokens and *2 layers*. Interestingly, the general structure of the feasible space remains very close! This means that at least in this setting, our analysis (to our surprise) seems to hold also for multiple layers. We hypothesize that functionality is distributed across the layers, with each layer performing operations similar to those of a single layer, resulting in minimal additional model capacity when extending to multiple layers. A manual analysis of the layers function would be very interesting. It does however seem that the results are slightly better on average -- possibly showing that overparameterization in terms of #layers is beneficial for training. Thanks for suggesting this nice experiment, we will continue to look into this and we added the results to the appendix D.4 of the paper!
>
> > Have you done initial strategies for solving other tasks, such as look-up and sorting?
>
> Indeed, our initial experiments were conducted over a wider range of tasks including look-up and sorting, mostly inspired from RASP. We saw that for look-up the dot-product attention mechanism is required for a single layer -- but there is not a similarly complex and perhaps counterintuitive behavior over varying architectures we observed for the histogram task. For the look-up linear fails to learn anything. For sorting, a linear layer with enough hidden neurons is actually enough to learn how to sort, but crucially the number of tokens cannot be exceedingly high. We found this extremely surprising, but in the end the mechanism used seems to be a linear layer which encodes a position and creates a bag of the tokens in the new mixed token -- the feedforward layer is then capable of identifying a position $i$ from the mixed token and to then extract the $i$’th largest token from the representation of the whole sequence. To us the exact mechanism remains yet elusive.
>
> If you have some **follow-up comments**, time and interest to discuss the questions or more points on the relevance of our work, we are happy to engage further!
> Likewise, if your concerns were addressed sufficiently, we would appreciate it if you would consider raising your score.

---

> > ### Comment · Reviewer_yFeG · 2024-11-22
> >
> > Thank you for further clarifying your selection of the histogram task and the extra experiment on the two-layer transformers. It is nice to see the structure holds up in the two-layer setting as it signals these findings can be extrapolated to more complicated, real-world settings. I am increasing my score to 6.

---

### Official Review · Reviewer_sVBS · 2024-11-04

**Soundness:** 3
**Presentation:** 3
**Contribution:** 2
**Rating:** 6
**Confidence:** 2

**Summary:**

This paper explores the performance of small Transformer models in performing counting tasks (specifically histogram tasks), analyzing how the collaboration of different components in the model architecture affects the final solution.
The authors found that in the histogram task, the predictive performance of the model is closely related to the vocabulary size, embedding dimension, token mixing mechanism and the capacity of the feed-forward block. Small Transformers can implement two counting strategies, the choice of which is influenced by hyperparameters and synergies between components. Even small architectural adjustments can lead to significant changes in the solutions learned by the model.

**Strengths:**

1. This paper demonstrates how various components of the transformer affect the model's learning ability, which can help understand the internal mechanisms of  transformer

2. The authors provide a simple yet effective testing environment, making it possible to evaluate the impact of the model architecture on solutions more clearly.

3. The visualizations are well done, making it easier to understand.

**Weaknesses:**

1. The paper analyzes only a simple single-layer transformer structure, while scaled-up models in real applications may exhibit different behaviors. Additionally, this paper overlooks the impact of the autoregressive mechanism and positional encoding.

2. Although the paper reveals the impact of architectural details on model performance, it lacks clear guidance for practical applications. For instance, it analyzes how relation-based counting, inventory-based counting, and softmax significantly influence the model's learning strategies and performance. However, I find it challenging to relate these methods to transformer models in real-world applications. I believe that adjustments in model design often need to be validated in practical settings, yet the paper does not provide specific application examples or optimization suggestions.

**Questions:**

Could you explain in detail what the "online learning setting" in Line 153 means?

---

> ### Author Response · Authors · 2024-11-19
>
> Dear reviewer sVBS,
>
> We thank you for appreciating the histogram task as an effective framework to test the model internals. We are happy to hear that the visualizations aided your understanding, and that from your point of view the histogram task is a meaningful example demonstrating how subtle architectural choices influence the learned algorithm.
>
> Since some of your concerns have been raised by other reviewers, we provide a more detailed response in a **shared reply**. We provide more specific details regarding the points you raised in the following.
>
> **Generalization to larger models:** We agree that real-world large-scale models are likely to exhibit even more complex behaviors than the toy example discussed in our work, particularly given the influence of autoregressive mechanism and positional encoding on the learned algorithms. Directly studying these models, especially from a theoretical perspective is a very hard challenge.
> However, we believe this complexity underscores the value of our approach. Even in our simplified setting, we observed an unexpectedly rich phenomenology (which we believe is interesting in and of itself regardless of its implications on larger models), demonstrating that even seemingly straightforward models can exhibit complex behaviors that deserve rigorous investigation. This suggests that simpler models, while not fully representative of large-scale transformers, can still serve as valuable tools for building intuition about the mechanisms at play in more complex architectures.
>
> As also detailed in the general response, several open questions remain in the characterization of these models. In particular, the role of the softmax operator, the interplay between their components and the impact of their hyperparameters are often investigated from an empirical perspective. In our work, similarly to many other research works focusing on toy tasks, we precisely address these questions in a restricted and fully controlled setup by providing theoretical constructions that match empirical observations.
>
> **Guidance for practical applications:** Regarding the lack of specific adjustments of real models, that you understandably would like to see: we find the histogram task valuable as an initial step toward linking a detailed understanding of small models with practical applications. Solving the histogram task inherently involves *selection* (of equal tokens), *aggregation* (into a single vector), and *summarization* (extracting the count) —key capabilities that are central to many real-world applications, such as addressing information overload, one of the primary use cases of modern large language models.
> While reasoning tasks—such as those explored through in-context learning—have served as useful toy testbeds inspiring extensive research, the histogram task can be viewed as an example for studying summarization. We hope that insights gained from the histogram task can, in future work, guide the analysis of real multi-layer models, particularly in exploring the softmax’s role in selecting inputs or handling tokens like the BOS (beginning of sequence) and distinguishing between inventory-based and relation-based information extraction.
>
> Other than constructing a parallel between the task and real applications, the simple vector arithmetic we used in the theoretical constructions can be instructive. Even in larger models, representations are often linear [https://arxiv.org/abs/2311.03658](https://arxiv.org/abs/2311.03658), but not all representations can be orthogonal due to the typical mismatch between the number of tokens and embedding dimensions. It is interesting to understand how these entangled, non-orthogonal embeddings interact to advance our understanding of relationships between representations within larger networks.
>
> **Online Learning Setting:** Regarding your question, the online learning setting in line 153 refers to the fact that during learning with Adam, at every step we sample a fresh batch of samples from our generative model of the histogram task. We clarify this in a footnote of the updated version.
>
> If you have **more concerns and comments** you would like to discuss, we would be happy to hear more from you. If your concerns were addressed sufficiently, we would appreciate it if you would consider raising your score.

---

> > ### Author Response · Authors · 2024-11-27
> >
> > Dear reviewer sVBS,
> >
> > As the discussion phase nears its end, we would still be curious to hear your feedback on our reply -- if your time allows.
> >
> > We did do our best to address your concerns, but in case you need more information to make your final evaluation, we remain at your disposal.
> >
> > Best!

---

> > > ### Comment · Reviewer_sVBS · 2024-11-30
> > > **Reviewer Feedback**
> > >
> > > I appreciate the author's response, and I decide to raise my score to 6.

---

### Official Review · Reviewer_jaXn · 2024-11-06

**Soundness:** 2
**Presentation:** 2
**Contribution:** 2
**Rating:** 5
**Confidence:** 3

**Summary:**

In this paper, the author studied the relationship between attention and the feedforward layer through the task of histogram task.

**Strengths:**

The author studied an interesting topic about the relationship between the attention and the feedforward layer.

**Weaknesses:**

1. The paper’s narrative could use some improvement for better clarity and flow.

2. This paper studies the relationships among components in transformers within the context of histogram tasks. However, there isn’t enough evidence to show that this empirical study on histograms alone can adequately represent the properties of transformers in general. The cited work, Thinking like Transformers, also uses histograms but includes reverse sorting and other tasks to provide a broader context.

3. Another concern is the significance of these findings—why are they important, and do they provide meaningful insights?

4. There’s little discussion on how these findings could guide the adjustment or application of transformer structures in practical contexts.

**Questions:**

See weakness.

---

> ### Author Response · Authors · 2024-11-19
>
> Dear reviewer jaXn,
>
> We thank you for taking the time to review our work, and for recognizing the value of our analysis on the interaction between feed-forward and attention layers in the histogram task.
>
> **Paper clarity and flow:** We are sorry to hear that you find that our narrative requires more clarity. Could you please provide us some concrete feedback on which parts were especially confusing to you? As all the other reviewers highlighted that generally the narrative was easy to follow and logical, your feedback specifically would be important for us to further improve the paper.
>
> **Significance and outreach:** Regarding your broader concerns for the contribution of our work, which are shared with some of the other reviewers, we address them in a common reply above. A discussion more specific to your concerns follows.
>
> **Limited scope of the histogram task:** We agree that the histogram task alone cannot capture the full complexity of the mechanisms arising in large scale modern transformers. Providing a rigorous theoretical analysis of such models is very hard given the complex interdependencies between real-world datasets, architectural components and hyperparameters. However, given the widespread use of transformers and the lack of a clear understanding of their inner mechanisms even in relatively simple cases, our choice of the histogram task and simple one-layer transformer blocks provides a first step towards understanding some of the properties of these models. We believe these kinds of analyses are very important in general as many advances in modern deep learning are driven by heuristics choices, not always adequately motivated even in very simple settings. To this extent, the histogram task represents a seemingly simple test ground which nevertheless exhibits a very rich phenomenology. It allows us to highlight the impact of different architectural design choices and hyper parameters, often overlooked in most empirical studies. Beyond offering an interesting playground to study different components, the histogram task presents some properties that are also shared by more complex tasks, such as summarisation and aggregation of information more generally.
>
> In addition, we would like to remark that there is a very fruitful line of work studying the properties of transformers applied to individual synthetic algorithmic tasks. For example, training small transformers to solve simple tasks like modular addition has led to the emergence of the grokking phenomenon, which is now the object of intense research efforts. Similarly to us [https://arxiv.org/abs/2402.03902](https://arxiv.org/abs/2402.03902) also considers the histogram task to unveil very peculiar phase transitions occurring in simple one-layer transformer models. Another example is the influential work [https://arxiv.org/abs/2306.17844](https://arxiv.org/abs/2306.17844) showing that simple one layer transformers can solve modular addition by implementing different algorithms exploiting the periodic structure of the task.
>
> **On RASP:** Thinking Like Transformers [https://arxiv.org/abs/2106.06981](https://arxiv.org/abs/2106.06981) is a seminal paper from which we indeed drew inspiration in designing our study. Our paper builds on this work and extends its analysis from a specific angle that we believe the original paper did not explore. More specifically, while the RASP language can be used to programmatically represent transformers, it may overlook the importance of the scale of the model and, in particular, its memorisation capacity.
>
> In our work, for example, we show that even models lacking algorithmic alignment with the task (e.g. Linear) can still manage to solve it by falling back to memorisation (inventory-based counting), provided $d$ and $p$ are large enough. In addition, in our answer to reviewer yFeG, we share some results from our preliminary explorations which investigated a broader range of tasks representable in the RASP language, which might be interesting to you as well.
>
> **On the soundness score:** Finally, we would appreciate a clarification on your low score on the soundness of our work - 2 - fair. We did our best to check our theoretical results and substantiate our claims with sufficient evidence from experiments and related work. Since your criticism mainly seems to focus on the relevance and direct application of our work to large transformers, it is not clear which parts of the manuscript concern you the most in terms of soundness. For this reason, we would greatly appreciate it if you could provide more detailed feedback, so that we can check again the points that lead you to your judgment and further improve the paper based on your indications.
>
> We hope our answers address your concern and could possibly lead you to raise the score. We would be grateful for any feedback you may give us that could improve the current version of the paper.

---

> > ### Comment · Reviewer_jaXn · 2024-11-26
> >
> > Thank you for answering my questions in your rebuttal. I will increase my score to 5.

---

### Author Response · Authors · 2024-11-19
**General Reply (1/2)**

We would like to thank the reviewers for taking their time to read our paper. We were pleased that all reviewers found our analysis interesting and the research questions we pose relevant. We have also appreciated that most reviewers found our work sound and well-written. We address criticism raised by multiple reviewers jointly below.

**Why the histogram task?**

Counting is a primitive operation and studying which circuit enables its execution in neural networks is an interesting problem in and of itself. The histogram task, as a simple counting task, serves as an effective test bench to explore the internal mechanisms of sequence-to-sequence models. Beyond counting, this task directly connects to key operations like selection, aggregation and summarization — these abstract processes are central to many practical applications of language models in everyday scenarios.

The primary question we address is understanding the mutual relationships between transformer’s components. In the context of modern transformers, where architecture design is largely guided by heuristics, tackling this question presents significant challenges. To make this problem tractable, we opted for a setting that is theoretically manageable yet exhibits sufficiently rich and non-trivial phenomenology to yield meaningful insights on common architectural components.

Indeed, despite its simplicity, the task has been adopted in the community, in particular to study transformers. As also mentioned by reviewer jaXn, [https://arxiv.org/abs/2106.06981](https://arxiv.org/abs/2106.06981) for instance, uses this task to study the RASP language. More recently, [https://arxiv.org/pdf/2402.03902](https://arxiv.org/pdf/2402.03902) used the histogram task to uncover a phase transition occurring in simple one-layer transformers. In [https://arxiv.org/pdf/2406.02585](https://arxiv.org/pdf/2406.02585) the task is used in a slightly different autoregressive form without considering the precise implications for the architecture. All these works show that the histogram task can hide highly non-trivial phenomena whose understanding can provide some profound insights into the inner mechanisms of transformers.

---

> ### Author Response · Authors · 2024-11-19
> **General Reply (2/2)**
>
> **Significance and lack of generalizability to more realistic settings**
>
> We acknowledge that our study focuses on a simplified scenario, which may not immediately translate to all real-world applications. However, the primary aim of our work is to isolate and examine certain distinctive properties of transformer blocks that are challenging to identify in large-scale models.
> As acknowledged by reviewer sVBS, our work provides a "clear and simple testing environment, making it possible to evaluate the impact of the model architecture more clearly." We fully align with this perspective, as we believe that the problems explored in this study are also pertinent to understanding and analyzing more complex models.
>
> For example, **the role of the feedforward layers** and its relationship with the attention mechanism is often overlooked in the literature. Some empirical studies suggest that in transformers one can store factual associations in their weight matrices (e.g. [https://arxiv.org/pdf/2306.00802](https://arxiv.org/pdf/2306.00802) and [https://openreview.net/pdf?id=hwSmPOAmhk](https://openreview.net/pdf?id=hwSmPOAmhk)), effectively working like memories. In our setup, we show that the role of MLPs is tightly interconnected with the choice of the mixing mechanism and that indeed, in some cases, their role is very much aligned to that of a lookup-table (a.k.a. inventory).
>
> Our work also studies the **role of the softmax operator**. This component is the subject of many research works and its role is often the object of intense debate. It is not indeed clear whether this component is actually needed in the first place and what practical advantage it brings. Our work addresses this point  in a controllable setting from the perspective of the impact this component has on the hypothesis space of the model. We precisely show that the question does not have a straightforward general answer and that surprisingly the softmax can be both detrimental and beneficial depending on the case. Interestingly, the softmax can be helpful when the embedding dimension is smaller than the vocabulary size, a scenario which is the standard in real-world scenarios.
>
> Additionally, another important question concerns the **choice of hyperparameters** such as the embedding size and the hidden layer size of the MLP. These quantities have a significant impact on the final parameter count of the model and their choice is typically driven by heuristics. Our study shows that the choice of these hyperparameters is tightly connected with the architectural design of the model, i.e. the choice of its components, and highlights the importance of principled decisions in this context.
>
> Finally, while more complex models could lead to more intricate interdependencies between the components, it seems plausible that similar **vector arithmetic** to that employed in our proofs could emerge in subspaces of larger transformers. The linear representation hypothesis [https://arxiv.org/abs/2311.03658](https://arxiv.org/abs/2311.03658) states that representations in neural networks tend to be linear. At the same time not all representations can be orthogonal to each other since the number of tokens is typically much larger than the embedding dimensions. Understanding how such entangled non-orthogonal representations can be processed is therefore crucial to advance our knowledge of representations in relation to each other in larger networks.
>
> In conclusion, while our setting is simple and controllable, it reveals a rich and complex phenomenology that we can explain theoretically. This highlights the potential pitfalls of relying solely on intuition and heuristics in large-scale models, as even in seemingly straightforward scenarios like ours, we observe **counterintuitive behaviors that challenge conventional assumptions**. In our particular case, we plan to use our newly gained intuitions to investigate how the architectural components enable factual and text summarization in larger networks.
>
> The approach of using such a theoretical testbed to isolate given mechanisms -- here the highly non-trivial interaction between components in a scenario that isolates a recurring combination of tasks --  is quite common in the literature. It has often resulted in unveiling intriguing phenomena that would have been otherwise **overlooked by exclusively focusing on large-scale models** (e.g. [https://arxiv.org/pdf/2201.02177](https://arxiv.org/pdf/2201.02177), [https://arxiv.org/pdf/2301.05217](https://arxiv.org/pdf/2301.05217),[https://arxiv.org/pdf/2306.00802](https://arxiv.org/pdf/2306.00802),[https://arxiv.org/pdf/2402.03902](https://arxiv.org/pdf/2402.03902)). While not immediately applicable to real-world scenarios, they may help identifying new phenomena of intrinsic interest or to later contribute to the understanding of more complex models.

---

### Meta-Review · Area_Chair_YxNB · 2024-12-19

**Metareview:**

The paper investigates how different components of the transformer architecture, e.g. feedforward and attention layers, affect performance in counting tasks, specifically a histogram task in this paper. The reviewers appreciated the analysis of the learning ability in simple to understand scenarios and appreciated that the paper contains well done visualizations and structure. However, there were several shared concerns on i) a potentially limited analysis given the focus on single-layer transformer architectures, ii) limited discussion on how to leverage insights and lack of clear guidance for practical utility, iii) a too narrow investigation of just the histogram task.
The authors have mainly provided clarifications in the discussion period, many of which are textual in nature, and the updates made to the paper seem to be rather limited (there seems to only be a single sentence that changed). Respectively, some reviewers have raised their score slightly, whereas others remained slightly less convinced. Overall, the reviews all remain at borderline ratings after the discussion period. The AC believes that some of the aspects discussed and outlined are initial good steps to further improve the paper. However, the AC also agrees with the sentiment that the paper would benefit from thoroughly addressing above three points in a revised paper version. The AC thus recommends to reject the paper in its present form.

**Additional Comments On Reviewer Discussion:**

The reviewers have provided various feedback points, but all seem to have raised a common set of required improvements at the core. A part of these highly overlapping concerns were responded to in a common author response, but precise improvements to the paper are largely yet to be conducted. The changes during the rebuttal period thus remain limited. Although some reviewers have raised their scores, opinions on paper acceptance remain borderline. Reviewer aS5m points out that the significance of the results and take-aways remain unclear. The AC agrees that the paper would benefit from a broader analysis and a clearer form of practical guidance to increase potential impact. The AC further advises to conduct changes to the paper, which remain unclear from the discussion.

---

### Decision · Program_Chairs · 2025-01-22

Reject